# Boosting LLM Reasoning via Spontaneous Self-Correction

**Xutong Zhao**[1,2]**, Tengyu Xu**[1]**, Xuewei Wang**[1]**, Zhengxing Chen**[1]**, Di Jin**[1]**, Liang Tan**[1]**,
Yen-Ting Lin**[1]**, Zishun Yu**[1]**, Zhuokai Zhao**[1]**, Yun He**[1]**, Sinong Wang**[1]**, Han Fang**[1]**,
Sarath Chandar**[2]**, Chen Zhu**[1]

[1]MetaAI, [2]Mila - Quebec AI Institute & Polytechnique Montréal
{xutong.zhao@mila.quebec, zhuchen.eric@gmail.com}

## Abstract

While large language models (LLMs) have demonstrated remarkable success on a broad range of tasks, math reasoning remains a challenging one. One of the approaches for improving math reasoning is self-correction, which designs self-improving loops to let the model correct its own mistakes. However, existing self-correction approaches treat corrections as standalone post-generation refinements, relying on extra prompt and system designs to elicit self-corrections, instead of performing real-time, spontaneous self-corrections in a single pass. To address this, we propose **SPOC**, a *spontaneous self-correction* approach that enables LLMs to generate interleaved solutions and verifications in a *single inference pass*, with generation dynamically terminated based on verification outcomes, thereby effectively scaling inference time compute. SPOC considers a multi-agent perspective by assigning dual roles – solution proposer and verifier – to the same model. We adopt a simple yet effective approach to generate synthetic data for fine-tuning, enabling the model to develop capabilities for self-verification and multi-agent collaboration. We further improve its solution proposal and verification accuracy through online reinforcement learning. Experiments on mathematical reasoning benchmarks show that SPOC significantly improves performance. Notably, SPOC boosts the accuracy of Llama-3.1-8B and 70B Instruct models, achieving absolute gains of 8.8% and 11.6% on MATH500, 10.0% and 20.0% on AMC23, and 3.3% and 6.7% on AIME24, respectively.

## 1 Introduction

Large Language Models (LLMs) have showcased promising results across a broad spectrum of text generation tasks. Among the various domains of LLM applications, mathematical reasoning remains particularly challenging due to its symbolic and structured nature (Shao et al., 2024; Chen et al., 2024). Recent advances in self-correction (Shinn et al., 2023; Madaan et al., 2023) have emerged as a promising paradigm towards self-improvement through iterative critique and refinement of model's own responses.

However, the effectiveness and practicality of existing self-correction approaches remain unclear. Naive prompting methods may lead to minimal improvement or performance degradation without access to external feedback (Huang et al., 2023; Qu et al., 2024). Finetuning-based methods seek to address such issues by post-training the LLM on refinement data collected from oracles (Saunders et al., 2022; Qu et al., 2024) or the learner model itself (Kumar et al., 2024). Nonetheless, these approaches typically rely on a specific prompt after each model response to trigger self-reflection or correction (Figures 1a and 1b), necessitating additional system design to inject these prompts during inference. In other words, existing approaches lack the ability to spontaneously and adaptively self reflect and correct, resulting in ineffective test-time compute scaling and inflexible deployment in practice.

To address these challenges, we introduce SPOC, a *spontaneous self-correction* approach that enables LLMs to spontaneously generate interleaved solutions and verifications in a *single inference pass*. SPOC employs an open-loop inference paradigm, which triggers

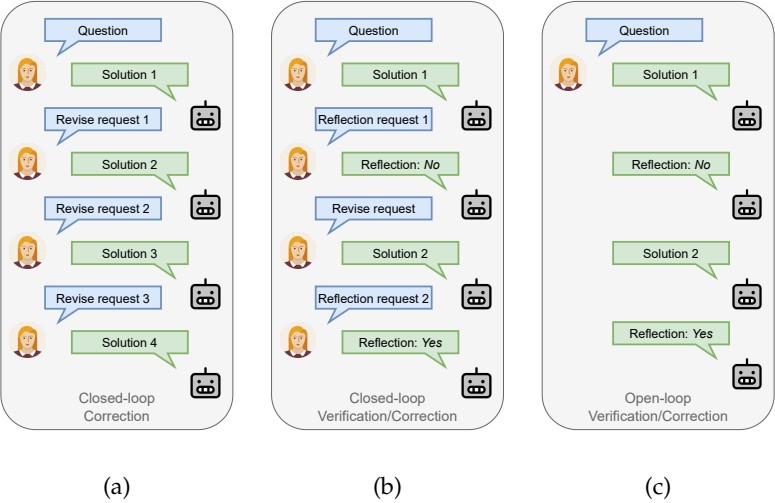

Figure 1: Multi-turn generation formalisms. (a) & (b) Sample closed-loop paradigms that require extra system designs and prompting to trigger and terminate correction; (c) Sample open-loop paradigm that spontaneously adapts generations.

self-correction only when the self-verification identifies errors, and iteratively revises the solution until it passes self-verification, without requiring any external interventions during response generation. It dynamically elicits and terminates generations on-the-fly using solely the model's inherent capabilities, thereby effectively scaling inference time compute. We consider a multi-agent formalism that models the alternating solutions and verifications as the interaction between a solution proposer and a verifier, and adopt a self-play training strategy by assigning dual roles to the same model. We adopt a simple yet effective approach to generate synthetic data from the initial model for supervised fine-tuning (Welleck et al., 2022), enabling the model to adhere to the multi-turn generation style, meanwhile developing capabilities for self-verification and inter-agent collaboration without distilling from a stronger teacher. We further boost the model's accuracy in its solution proposal and verification via online reinforcement learning, using the correctness of solutions and verifications as the reward.

Our main contributions are threefold:

- We demonstrate that generating self-verification and correction trajectories from the initial model's correct and incorrect outputs effectively bootstraps its spontaneous self-verification and correction behavior. We call out the importance of data balancing in achieving high verification accuracy in this stage, which in turn benefits the subsequent RL phase.

- We propose the message-wise online RL framework for SPOC, and present the formulation of RAFT (Dong et al., 2023) and RLOO (Ahmadian et al., 2024) as the RL stage of SPOC for enhancing self-verification and correction accuracies. Our results show that RLOO, augmented with process rewards for each solution or verification step, yields stronger results.

- We achieve significant improvements on math reasoning tasks across model sizes and task difficulties using our pipeline without distilling from stronger models. SPOC boosts the pass@1 accuracy of Llama-3.1-8B and 70B Instruct models—improving performance by 8.8% and 11.6% on MATH500, by 10.0% and 20.0% on AMC23, and by 3.3% and 6.7% on AIME24.

## 2 Related work

**Self-correction.** Given that high-quality external feedback is often unavailable across various realistic circumstances, it is beneficial to enable an LLM to correct its initial responses based on solely on its inherent capabilities. Prior works on such intrinsic self-correction (Huang et al., 2023) or self-refinement can be categorized into two groups based on the problem settings and correction mechanisms: prompting and finetuning. Recent works (Huang et al., 2023; Qu et al., 2024) show that prior prompting methods lead to minimal improvement or degrading performance without strong assumptions on problem settings. For instance, Shinn et al. (2023) rely on oracle labels which are often unavailable in real-world applications; Madaan et al. (2023) use less informative prompts for initial responses, resulting in overestimation of correction performance. Finetuning methods seek to improve correction performance via finetuning the LLM on refinement data, collected from human annotators (Saunders et al., 2022), stronger models (Qu et al., 2024), or the learner model itself (Kumar et al., 2024). However, these works lack the mechanisms that correct errors while generating solutions in a single inference pass (Ye et al., 2024). Our work is akin to concurrent works on self-correction (Ma et al., 2025; Xiong et al., 2025). Differently, Xiong et al. (2023) re-attempts a solution within the verification instead of evaluating the previous one; moreover, they only apply RAFT in their learning framework, while we also conduct experiments on RLOO. Ma et al. (2025) uses the more complex GRPO as their RL algorithm, while we show that better performance can be achieved in the same setting (Llama 3.1 8B) by using simpler RL algorithms like RAFT for SPOC.

**Multi-agent frameworks.** By introducing multiple roles into problem-solving, multi-agent formalisms serve as a different perspective to address complex reasoning tasks. Auto-Gen (Wu et al., 2023) and debate-based frameworks (Du et al., 2023; Liang et al., 2023) solve math problems through customized inter-agent conversations. Despite increased test-time computation, these works lack post-training for different agent roles, which may result in suboptimal performance or distribution shifts at inference time (Xiang et al., 2025). While other works train separate models to perform correction (Motwani et al., 2024; Havrilla et al., 2024; Akyürek et al., 2023; Paul et al., 2023), models do not perform spontaneous corrections during solution generations; instead, they require extra system designs to trigger and stop corrections at deployment. In contrast, our method enables dynamic inference-time scaling by improving the model's own *inherent* deliberation capabilities.

## 3 Method

In this section, we first introduce the multi-turn formalism, in which the agent performs interleaved solution and verification turns. We then discuss how we finetune the agent to ensure it consistently adheres to the multi-turn response style. We finally describe our online reinforcement learning scheme which further boosts the final accuracy of the policy. Figure 2 illustrates the two stages, fine-tuning and online RL, of SPOC.

### 3.1 Multi-turn formalism

**Problem setup.** Let $\mathcal{D} \equiv \mathcal{X} \times \mathcal{Y} = \{(x_i, y_i^*)\}_{i=1}^N$ be a dataset of $N$ math problems, where each pair $(x, y^*)$ contains a question $x_i$ and the corresponding solution $y_i^*$ with ground-truth final answer. An LLM agent is defined by the policy $\pi_\theta(\cdot|x)$, parameterized by $\theta$, that generates the solution $y$ to solve the given problem $x$.

**Alternated-turn generation.** Suppose given a question $x$, the LLM generates a trajectory consisting of $L$ interleaved solutions and verifications $\tau = (y_1, v_1, \ldots, y_L, v_L)$, where a solution $y_l$ indicating the model's $l$-th complete solution attempt that reaches a final answer, and a verification $v_l$ indicating the $l$-th self-verification validating correctness of the solution $y_l$. For clarity, message or turn refers to each single solution $y_l$ or verification $v_l$, and response or generation $\tau$ refers to the entire trajectory until the end. For brevity, we denote

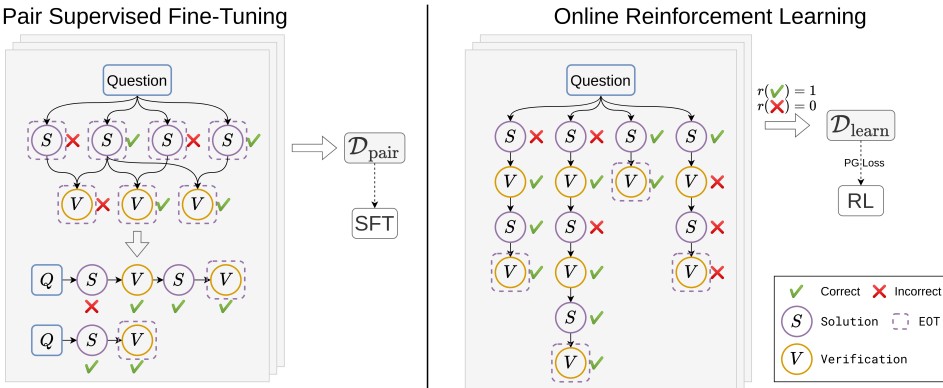

Figure 2: SPOC training overview. Left: PairSFT for initializing multi-turn generation. Right: Online RL for policy optimization.

previous $l$ turns by: $\tau_l = (y_{1:l}, v_{1:l})$ and $\tau_l^{\text{vf}} = (y_{1:l}, v_{1:l-1})$. The timestep $t \in \mathbb{N}_0$ indicates a single decoding step where the LLM outputs one token from its policy distribution.

**Multi-agent formulation.** We model the reasoning task as an extensive-form game (EFG) (Osborne, 1994; Shoham & Leyton-Brown, 2008), which generalizes the Markov Decision Process (MDP) (Sutton, 2018) to a turn-taking interaction between solution proposer and verifier. At each turn, the proposer outputs a solution to the given math problem, and the verifier assesses its correctness. In this context, the EFG is a tuple $\langle \mathcal{N}, \mathcal{A}, \mathcal{S}, \mathcal{T}, r, \mathcal{I}, \gamma \rangle$, where $\mathcal{N} = \{1, \ldots, n\}$ is the set of $n = 2$ players (i.e. the proposer and verifier), $\mathcal{A}$ is a finite set of actions (i.e. the LLM's token space), $\mathcal{S}$ is a finite set of states (i.e. each state is a question and a sequence of reasoning/verification steps in context), $\mathcal{T} \subset \mathcal{S}$ is a subset of terminal states (i.e. complete response trajectories $\tau = (y_1, v_1, \ldots, y_L, v_L)$), $r : \mathcal{T} \times \mathbb{N}_0 \to \Delta_r^n \subset \mathbb{R}^n$ is the reward function assigning each player a scalar utility at terminal states (i.e. $\Delta_r = \{0, 1\}$ characterizes binary outcome feedback), $\mathcal{I} : \mathcal{S} \to \mathcal{N}$ is a player identity function identifying which player acts at $s$ (i.e. $\mathcal{I}(\tau_l) = 1$ and $\mathcal{I}(\tau_l^{\text{vf}}) = 2$), and $\gamma \in [0, 1]$ is the discount factor.

Unlike the general definition of EFGs, we do not distinguish between histories and states due to the deterministic dynamics and perfect-information nature in mathematical reasoning (i.e. $\tau_{l+1} = \tau_l \cup \{y_{l+1}, v_{l+1}\}$). We denote the proposer's and the verifier's action spaces as $\mathcal{A}^{\text{sl}} \subset \mathcal{A}$ and $\mathcal{A}^{\text{vf}} \subset \mathcal{A}$, representing the set of solution and verification messages, respectively. We define a per-step reward function for a transition as $r(s, a)$ representing a vector of reward to both agents. The return for player $i \in \mathcal{N}$ is defined as $G_{t,i} = \sum_{k=0}^{\infty} \gamma^k r_i(s_{t+k}, a_{t+k})$. The corresponding state-action value function under policy $\pi$ is $Q_{\pi_i}(s, a) = \mathbb{E}_\pi[G_{t,i}|s_t = s, a_t = a]$.

To improve reasoning capabilities by learning from both solution and verification experiences, we adopt the commonly-used self-play strategy with parameter sharing (Albrecht et al., 2024), where the proposer policy $\pi^{\text{sl}} : \mathcal{S} \to \Delta(\mathcal{A}^{\text{sl}})$ and the verifier policy $\pi^{\text{vf}} : \mathcal{S} \to \Delta(\mathcal{A}^{\text{vf}})$ share the same set of parameters $\theta$. The policy $\pi_\theta$ outputs alternated solution and verification messages depending on the context[1].

**Policy optimization.** We optimize the policy $\pi_\theta$ by maximizing the KL-regularized learning objective

$$J(\theta) = \mathbb{E}_{s \sim \rho, a \sim \pi}[Q_\pi(s, a)] - \eta \cdot \mathbb{E}_{s \sim \rho}[\text{KL}(\pi_\theta | \pi_{\theta_0})] \tag{1}$$

where $\rho$ indicates the discounted state distribution, $\eta > 0$ is the KL-regularization coefficient, and $\pi_{\theta_0}$ is the reference policy parameterized by the initial parameters $\theta_0$. This objective has

---

[1]Different from the classic self-play in zero-sum games (e.g., AlphaZero (Silver et al., 2017)), ours involves non-symmetrical roles in the sense that two policies are different conditioned on the context.

| vf
sl | C | I |
|---|---|---|
| C | 1, 1 | 1, 0 |
| I | 0, 1 | 0, 0 |

(a) *Corr*

| vf
sl | C | I |
|---|---|---|
| C | 1, 0 | 1, 0 |
| I | 0, 0 | 0, 0 |

(b) *Last*

| vf
sl | C | I |
|---|---|---|
| C | 1, 1 | 1, 1 |
| I | 0, 0 | 0, 0 |

(c) *All*

Figure 3: Reward configurations for policy optimization, where sl, vf, C, I indicate solution, verification, correct, and incorrect, respectively. For *Last* and *All*, SPOC optimizes correct solutions (first row in each table) only when the last solution is correct.

a close-form solution for the optimal policy $\pi^*(a|s) = \frac{1}{Z(s)}\pi_{\theta_0}(a|s)\exp(\frac{1}{\eta}Q(s,a))$, where $Z(s) = \mathbb{E}_{a\sim\pi_{\theta_0}(\cdot|s)}[\exp(\frac{1}{\eta}Q(s,a))]$. Given our multi-agent formulation, this objective introduces an individual objective for each role, namely

$$J^{\text{sl}}(\theta) = \mathbb{E}[Q_\pi^{\text{sl}}(s,a)] - \eta^{\text{sl}} \cdot \mathbb{E}[\text{KL}(\pi_\theta^{\text{sl}}(\cdot|s)|\pi_{\theta_0}^{\text{sl}}(\cdot|s))] \tag{2}$$

$$J^{\text{vf}}(\theta) = \mathbb{E}[Q_\pi^{\text{vf}}(s,a)] - \eta^{\text{vf}} \cdot \mathbb{E}[\text{KL}(\pi_\theta^{\text{vf}}(\cdot|s)|\pi_{\theta_0}^{\text{vf}}(\cdot|s))] \tag{3}$$

Due to shared parameters across both roles, we jointly optimize both objectives using common generated trajectory experiences. Hence the optimal proposer and verifier policies satisfy $\pi^{\text{sl}*}(a|s) \propto \pi_{\theta_0}^{\text{sl}}(a|s)\exp(\frac{1}{\eta}Q^{\text{sl}}(s,a))$ and $\pi^{\text{vf}*}(a|s) \propto \pi_{\theta_0}^{\text{vf}}(a|s)\exp(\frac{1}{\eta}Q^{\text{vf}}(s,a))$, respectively, implying the optimal shared policy increases the probability of outputting high-rewarding solutions/verifications. Note that the optimal policy for the unregularized learning objective ($\eta = 0$) results in the maximizer of the action-value function: $\pi^*(\cdot|s) = \arg\max_{\pi\in\Delta(\mathcal{A})} \mathbb{E}_{a\sim\pi}[Q_\pi(s,a)]$, also yielding high probablity of generating high-rewarding messages.

**Reward setting.** To obtain a reward signal for each token in each message, we evaluate the outcome correctness of each message. In particular, we assume access to a rule-based checker for the final answer in the solution, and provide a binary outcome reward denoted by $r^{\text{sl}}(y,y^*) \in \{0,1\}$, where $r^{\text{sl}}(y,y^*) = 1$ when the model answer matches the ground-truth answer. Similarly, we parse the Yes/No conclusion in each verification, and denote the reward function by $r^{\text{vf}}(v,v^*) \in \{0,1\}$, with $v^* = r^{\text{sl}}(y,y^*)$ indicating the ground-truth verification. Figure 3a shows the joint reward setting, denoted by *Corr* hereafter. To obtain maximal returns against each other role, our reward setting admits one unique Nash equilibrium (Shoham & Leyton-Brown, 2008) with the joint policy (i.e. the shared policy $\pi$) generating both correct solutions and correct verifications.

## 3.2 Enabling multi-turn generation

Since off-the-shelf LLMs do not adhere to the response style of interleaved solution and verification turns by default, before conducting RL optimization, we first perform an initial finetuning with multi-turn data to enable such behaviour. To collect such data, we implement a variant of Pair-SFT (Kumar et al., 2024; Welleck et al., 2022) to construct synthetic correction responses.

In particular, we rollout the base policy $\pi_{\theta_0}$ to collect single-turn responses for each question $x_i \in \mathcal{X}$, denoted by $\{y_i^k\}_{k=1}^K \sim \pi_{\theta_0}(\cdot|x_i)$. For each response, we record its binary correctness using the solution reward function $r_i^k = r^{\text{sl}}(y_i^k, y_i^*)$. We obtain the verification message of one single-turn response by pairing it with a correct sampled response. To generate verification of one response, either correct or incorrect, we prompt the same base model $\pi_{\theta_0}$ to identify the potential error, briefly explain it, and output a final binary conclusion indicating correctness of the given solution. The entire verification message is denoted as $v_i \sim \pi_{\theta_0}(\cdot|x_i, y_i, y_i^*)$, where $y_i^*$ indicates the correct sample. We denote this synthetic multi-turn correction dataset as the Pair-SFT dataset $\mathcal{D}_{\text{pair}} = \{(x_i, y_i^-, v_i^-, y_i^*)\} \cup \{(x_i, y_i^+, v_i^+)\}$, where the $+/-$ superscripts indicates correctness of the corresponding solution turn. We

perform SFT finetuning on the base model, with tokens in incorrect messages masked out, and denote the finetuned model by $\pi_{\theta_{\text{sft}}}$. In practice, we observe that reweighting the subsets $\{(x_i, y_i^-, v_i^-, y_i^*)\}$ and $\{(x_i, y_i^+, v_i^+)\}$ to approximately the same scale leads to a $\pi_{\theta_{\text{sft}}}$ with higher verification accuracy and more stable RL training afterwards. The complete training data collection procedure is detailed in Algorithm 2.

When generating the verification messages, we adapt the generative critic method (Zhang et al., 2024; Zheng et al., 2024) that prompts the model to respond with rationales before judging solution correctness, except that our variant concisely explains the error rather than performing a chain-of-thought (COT) analysis. Obtaining a strong COT verifier requires explicit training and it is out of scope of this work. Prompt templates for data construction are detailed in Appendix E.

---

**Algorithm 1** SPOC Message-wise Online Reinforcement Learning

1: **Inputs:** Question-answer dataset $\mathcal{D} = \mathcal{X} \times \mathcal{Y} = \{(x_j, y_j^*)\}_{j=1}^N$, policy model $\pi_\theta$ parameterized by $\theta$, number of questions $N$, number of steps $T$, number of rollouts per question $K$, batch size $B$, rule-based solution correctness reward function $r^{\text{sl}}(y, y^*) \in \{0, 1\}$, verification correctness reward function $r^{\text{vf}}(v, v^*) \in \{0, 1\}$
2: **for** $i = 1, \ldots, T$ **do**
3:     Sample a batch $\mathcal{D}_i \subset \mathcal{D}$ of size $B$
4:     Sample $K$ trajectories for each $x_j \in \mathcal{X}_i$: $\{\tau_j^k\}_{k=1}^K \sim \pi_\theta(\cdot|x_j)$, where $\tau_j^k = (y_{1:L_k}^{j,k}, v_{1:L_k}^{j,k})$
5:     Label binary rewards: $r_{j,k,l}^{\text{sl}} = r^{\text{sl}}(y_l^{j,k}, y_j^*), r_{j,k,l}^{\text{vf}} = r^{\text{vf}}(v_l^{j,k}, v_{j,k,l}^*)$, where $v_{j,k,l}^* = r_{j,k,l}^{\text{sl}}$
6:     Update policy with any policy optimization algorithm (e.g. Algorithm 3, Algorithm 4)
7: **end for**
8: **return** $\pi_\theta$

---

### 3.3 Online reinforcement learning

With the multi-turn problem formulated and the agent adhering to the multi-turn responses style, we conduct online reinforcement learning to improve the policy performance. The overall message-level RL training procedure is described in Algorithm 1. While SPOC is compatible with any policy optimization method, we apply RAFT (Dong et al., 2023) unless otherwise specified. The RAFT policy optimization algorithm is presented in Algorithm 3.

Besides the RAFT policy optimizer, we also implement an RLOO (Ahmadian et al., 2024) variant, which replaces the leave-one-out procedure with subtraction of the mean reward across all messages, followed by division by the standard deviation. We refer to this approach as RLOO for brevity. Unlike the best-of-N (BoN) response selection strategy in RAFT, RLOO optimizes the policy using all generated responses, enjoying better sample efficiency. The RLOO policy optimization is detailed in Algorithm 4.

## 4 Experiments

In this section we present empirical experiments on math reasoning benchmarks. We first overview the tasks we conduct experiments on. We then describe the experimental setup and evaluation protocols. Finally we discuss the results and provide ablation studies.

### 4.1 Experimental setup

**Tasks.** We perform experiments on established math reasoning benchmarks. To enable rule-based answer checking, all problems in selected benchmarks require a verifiable final output. We evaluate models on benchmarks: (1) MATH500 (Lightman et al., 2023), a curated dataset of 500 problems selected from the full MATH (Hendrycks et al., 2021) evaluation set; (2) AMC23 (AI-MO, 2023), a dataset of 40 challenging competition questions; (3) AIME24 (AI-MO, 2024), a dataset of 30 more difficult competition problems.

| Approach | MATH500 | AMC23 | AIME24 |
|---|---|---|---|
| Llama-3.1-8B-Instruct (Dubey et al., 2024) | 52.2 | 22.5 | 3.3 |
| SFT | 53.6 | 32.5 | 3.3 |
| RAFT | 55.2 | 27.5 | 6.7 |
| PairSFT | 53.8 | 22.5 | **10.0** |
| Self-Refine (w/o oracle) | 39.4 | 20.0 | 3.3 |
| Self-Refine (w/ oracle) | 57.0 | 35.0 | 3.3 |
| S²R-BI∗ (Ma et al., 2025) | 49.6 | 20.0 | **10.0** |
| S²R-PRL∗ | 53.6 | 25.0 | 6.7 |
| S²R-ORL∗ | 55.0 | **32.5** | 6.7 |
| SPOC | **61.0** | **32.5** | 6.7 |
| Llama-3.1-70B-Instruct (Dubey et al., 2024) | 65.8 | 32.5 | 16.7 |
| SFT | 70.4 | 45.0 | 13.3 |
| RAFT | 74.2 | 52.5 | 20.0 |
| PairSFT | 74.8 | 47.5 | 23.3 |
| Self-Refine (w/o oracle) | 54.2 | 42.5 | 13.3 |
| Self-Refine (w/ oracle) | 72.2 | 47.5 | 26.7 |
| SPOC | **77.4** | **52.5** | **23.3** |
| Llama-3.3-70B-Instruct (AI, 2024) | 75.6 | 57.5 | **26.7** |
| SFT | 73.6 | 55.0 | 23.3 |
| RAFT | 76.6 | 62.5 | 20.0 |
| PairSFT | 75.0 | 62.5 | 23.3 |
| Self-Refine (w/o oracle) | 75.4 | 60.0 | 33.3 |
| Self-Refine (w/ oracle) | 76.2 | 65.0 | 26.7 |
| SPOC | **77.8** | **70.0** | 23.3 |
| DeepSeek-R1-Distill-Llama-8B (Guo et al., 2025) | 62.6 | 62.5 | 26.7 |
| SFT | 76.8 | 65.0 | 30.0 |
| RAFT | 74.2 | 62.5 | 6.7 |
| PairSFT | 73.2 | 77.5 | 16.7 |
| Self-Refine (w/o oracle) | 67.4 | 75.0 | 10.0 |
| Self-Refine (w/ oracle) | 71.2 | 65.0 | 40.0 |
| SPOC | 77.6 | 70.0 | 23.3 |
| SPOC-RLOO | **87.2** | **87.5** | **50.0** |
| DeepSeek-R1-Distill-Llama-70B (Guo et al., 2025) | 82.8 | 72.5 | 60.0 |
| SFT | 90.6 | 80.0 | 40.0 |
| RAFT | 87.4 | 85.0 | 50.0 |
| PairSFT | 92.6 | **95.0** | 63.3 |
| Self-Refine (w/o oracle) | 86.2 | 80.0 | 30.0 |
| Self-Refine (w/ oracle) | 88.6 | 72.5 | 30.0 |
| SPOC | 89.6 | 85.0 | 53.3 |
| SPOC-RLOO | **94.6** | 92.5 | **76.7** |
| Gemini-1.5-Flash (4-shot)∗ (Team et al., 2024) | 54.9 | - | - |
| SCoRe∗ (Kumar et al., 2024) | 64.4 | - | - |
| Llama-3-8B-Instruct (4-shot)∗ (Meta, 2024) | 30.0 | - | - |
| Self-rewarding IFT∗ (Xiong et al., 2025) | 27.9 | - | - |
| Self-rewarding-IFT + Gold RM∗ | 33.9 | - | - |
| DeepSeek-R1-Distill-Llama-8B-R1tok-avg@4† | 88.9 | 92.5 | 48.3 |
| DeepSeek-R1-Distill-Llama-8B-R1tok† | 82.2 | 87.5 | 36.7 |
| DeepSeek-R1-Distill-Llama-8B∗ | 89.1 | - | 50.4 |
| DeepSeek-R1-Distill-Llama-70B-R1tok-avg@4† | 94.3 | 94.4 | 65.9 |
| DeepSeek-R1-Distill-Llama-70B-R1tok† | 91.2 | 80.0 | 56.7 |
| DeepSeek-R1-Distill-Llama-70B∗ | 94.5 | - | 70.0 |
| Qwen2.5-Math-7B-Instruct† (Yang et al., 2024) | 82.8 | 62.5 | 16.7 |
| Qwen2.5-Math-72B-Instruct† | 84.8 | 72.5 | 26.7 |
| O1∗ | 94.8 | - | 74.4 |
| GPT-4o∗ | 60.3 | - | 9.3 |
| Claude 3.5 Sonnet∗ | 78.0 | - | 16.0 |

Table 1: Main evaluation results. Bold and underlined performance scores indicate the best learning and best prompting (if any) results under each initial model, respectively. Blue means ours, and green means other RL based approaches. Baselines marked with ∗ are reported by the original works, while those marked with † are obtained by evaluating the open-source models on Huggingface using greedy decoding (unless stated otherwise). "R1tok" indicates the model is evaluated using the R1 modified tokenizer and chat configs. "avg@4" indicates the model is evaluated using sampling, with the temperature of 0.6, the top-p value of 0.95, and 4 responses generated per question to compute the mean pass@1 (Guo et al., 2025).

| Base Model trained w/ SPOC | Base.Acc. | Verif.Acc.@t1 | Acc.@t1 | Acc.@t2 | $\Delta(t1, t2)$ | $\Delta_{c \to i}$ | $\Delta_{i \to c}$ |
|---|---|---|---|---|---|---|---|
| Llama-3.1-8B-Instruct | 52.2 | 80.2 | 59.0 | 61.0 | 2.0 | 8/29 | 18/79 |
| Llama-3.1-70B-Instruct | 65.8 | 80.0 | 77.0 | 77.4 | 0.4 | 3/10 | 5/8 |
| Llama-3.3-70B-Instruct | 75.6 | 81.8 | 77.8 | 77.8 | 0.0 | 1/4 | 1/20 |

Table 2: Performance across first two solution turns on MATH500. $\Delta_{c \to i}$ and $\Delta_{i \to c}$ indicate $\frac{n_{c \to i}}{n_{c \to c} + n_{c \to i}}$ and $\frac{n_{i \to c}}{n_{i \to i} + n_{i \to c}}$ over turns, respectively.

**Evaluation protocol.** Our primary evaluation metric is the final answer accuracy. We additionally report cross-solution correction accuracy serving as a complementary evaluation.

For all experiments, we finetune Llama-3-Instruct models (Dubey et al., 2024) (3.1-8B & 70B, 3.3-70B, DeepSeek-R1-Distill-Llama 8B & 70B) as the base models. We conduct training using the NuminaMath dataset (LI et al., 2024), which consists of training sets from various data sources, covering a wide range of mathematical topics and difficulty levels. We exclude the Orca-Math dataset (Mitra et al., 2024) and synthetic data subset since their correctness are not human-validated despite their large scale.

For evaluations, we report the pass@1 accuracy of the final answer. We use greedy decoding and zero-shot COT prompting unless otherwise specified. As mentioned in previous sections, we do not utilize additional external instructions to prompt the finetuned model to attempt another solution trial; instead the model spontaneously performs self-verification to determine whether another attempt is needed. Our prompt templates for evaluation are included in Appendix E.

**Implementation details.** All models are prompted with the original Llama tokenizer and chat configs (Dubey et al., 2024) unless otherwise specified. All models except the DeepSeek-R1-Distill-Llama based ones are evaluated using the maximum generation length of $6,144$ tokens, while the DeepSeek-R1-Distill-Llama based models are evaluated using the maximum generation length of $32,768$ tokens, as per Guo et al. (2025). To support training with multi-message responses, we utilize different special termination tokens for each model message. In particular, in each model response each message starts with `assistant` header tokens, indicating the source of message is the model. Besides, every assitant message except the last ends with an `<|eom_id|>` termination token, representing the end of one message. The last assistant message ends with an `<|eot_id|>` token, which concludes the entire model response. We implement RAFT (Dong et al., 2023) under the CGPO (Xu et al., 2024) framework, which allows for filtering out prompts whose all corresponding sampled responses contain no correct solutions or verifications.

## 4.2 Results

Table 1 presents the comprehensive evaluation results, showing the comparisons across different initial models and parameter scales. In general, SPOC consistently outperforms the base models on all initialization models across all benchmark tasks. Notably, SPOC enhances the accuracy of Llama3.1 8B and 70B, reaching gains of 8.8% and 11.6% on MATH500, 10.0% and 20.0% on AMC23, and 3.3% and 6.7% on AIME24, respectively. This result highlights the effectiveness of SPOC across different parameter scales and task difficulties.

SPOC also achieves consistent enhancement when fine-tuned with strong initial models. Despite marginal improvement on Llama3.3-70B model, SPOC obtains significant overall outperformance compared to the baselines after finetuning the DeepSeek-R1-Distill-Llama models. Respectively on MATH500/AMC23/AIME24, SPOC reaches 77.6%/70.0%/23.3% with the 8B model, and 89.9%/85.0%/53.3% with the 70B model. Furthermore, SPOC achieves more drastic performance improvement using the RLOO policy optimizer, obtaining 87.2%/87.5%/50.0% with the 8B model, and 94.6%/92.5%/76.7% with the 70B model. It is important to note that the gap between our evaluation of DeepSeek-R1-Distill-Llama base models for post-training and their corresponding R1tok results is attributed to different tokenizers and chat configurations.

Table 2 shows performance across the first two solution turns on MATH500. Overall, SPOC achieves consistent improvement on the second solution turns over the first. With the smaller Llama3.1-8B model, SPOC shows more inclination to generate a second solution turn, resulting in a more significant improvement margin. With larger 70B models that achieve higher final accuracy, on the other hand, SPOC tends to get the first solution message correct in the first place, resulting in an already strong turn1 performance and a marginal $\Delta(t1, t2)$. Such behaviour is well aligned with our expected Nash equilibrium admitted by the *Corr* reward setting, where policy optimization encourages the joint policy to generate both correct solutions and correct verifications in the first place. The complete per-turn performance analysis and diagnostics of verifier reliability are presented in Appendix C.

Table 3 shows the performance of applying multiple iterations of PairSFT-RL training procedure. Results indicate that the second iteration still leads to overall consistent improvement over all models. Although the overall improvement is mainly marginal, the second iteration shows a larger gain in challenging competition benchmarks. For instance, with Llama3.1-70B, iter2 improves over iter1 by 10% and 6.7% on AMC23 and AIME24, respectively.

| Approach | MATH500 | AMC23 | AIME24 |
|---|---|---|---|
| Llama-3.1-8B-Instruct | 52.2 | 22.5 | 3.3 |
| PairSFT (iter1) | 53.8 | 22.5 | 10.0 |
| SPOC (iter1) | 61.0 | 32.5 | 6.7 |
| PairSFT (iter2) | 60.8 | 35.0 | 6.7 |
| SPOC (iter2) | 62.0 | 32.5 | 10.0 |
| Llama-3.1-70B-Instruct | 65.8 | 32.5 | 16.7 |
| PairSFT (iter1) | 74.8 | 47.5 | 23.3 |
| SPOC (iter1) | 77.4 | 52.5 | 23.3 |
| PairSFT (iter2) | 76.4 | 67.5 | 20.0 |
| SPOC (iter2) | 77.6 | 62.5 | 30.0 |
| Llama-3.3-70B-Instruct | 75.6 | 57.5 | 26.7 |
| PairSFT (iter1) | 75.0 | 62.5 | 23.3 |
| SPOC (iter1) | 77.8 | 70.0 | 23.3 |
| PairSFT (iter2) | 79.6 | 72.5 | 26.7 |
| SPOC (iter2) | 79.8 | 70.0 | 30.0 |

Table 3: Iterative training performance. The second iteration still yields overall consistent improvement across all models.

## 4.3 Ablations

We conduct ablation experiments on different reward configurations, as overviewed in Figure 3. We present comparisons with the default *Corr* reward setting in Table 4, using Llama-3.1-8B-Instruct as the base model. Compared to *Corr*, the ablation variants *Last* and *All* do not yield a unique Nash equilibrium; instead, they promote generating correct solutions regardless of the correctness of verifications. Results show that both variants still improve performance over the baseline; however, they both underperform *Corr* on two out of three tasks. *Last* and *All* obtains only one more correct answer than *Corr* in AIME24 and AMC23, respectively, while the performance discrepancy on MATH500 dominates the overall gap. The ablation highlights the importance of jointly optimizing the correctness of both solutions and verifications.

| Model | MATH500 | AMC23 | AIME24 |
|---|---|---|---|
| Base | 52.2 | 22.5 | 3.3 |
| SPOC-*Corr* | 61.0 | 32.5 | 6.7 |
| SPOC-*Last* | 59.8 | 27.5 | 10.0 |
| SPOC-*All* | 58.4 | 35.0 | 6.7 |

Table 4: Ablation experiments under different reward settings. Experiments are conducted on the Llama-3.1-8B-Instruct model.

# 5 Conclusions

In this work, we tackle the mathematical reasoning challenge for Large Language Models by promoting intrinsic self-corrections. We propose SPOC, a novel approach that enables spontaneous, real-time solution proposal and verification within a single inference pass. SPOC frames the reasoning process as a multi-agent collaboration, where the model assumes both the roles of a solution proposer and verifier. SPOC dynamically elicits and terminates reasoning generations based on verification results, which flexibly and efficiently scales inference-time compute while improving accuracy. SPOC leverages synthetic data for fine-tuning and further enhances performance via online reinforcement learning, without requiring human or oracle input. Comprehensive empirical evaluations on challenging math reasoning benchmarks showcase SPOC's efficacy, yielding substantial performance improvement.

Our results highlight the potential of spontaneous self-correction as an effective strategy for advancing LLM reasoning capabilities. To address the prohibitive length of long CoTs (Marjanović et al., 2025), future work could explore extending SPOC to partial solutions in long reasoning chains, using step-level process rewards to guide RL training and enable dynamic revisions when errors are detected until reaching the final answer. It would also be interesting to adopt SPOC to broader reasoning domains beyond mathematics, further enhancing its applicability.

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

# A Algorithms

---

**Algorithm 2** Pair-SFT Data Construction

---

1: **Inputs:** Question-answer dataset $\mathcal{D} = \mathcal{X} \times \mathcal{Y} = \{(x_i, y_i^*)\}_{i=1}^N$, policy model $\pi_0$, number of questions $N$, number of single-turn rollouts per question $K$, rule-based solution correctness reward function $r^{\text{sl}}(y, y^*) \in \{0, 1\}$, single-turn sampling set $\mathcal{D}_{\text{rjs}} \leftarrow \{\}$, multi-turn correction set $\mathcal{D}_{\text{pair}} \leftarrow \{\}$, verification message validator $f^{\text{vf}}(v) \in \{0, 1\}$
2: **for** $i = 1, \ldots, N$ **do**
3:     // Solution rollout
4:     Sample $K$ solutions for each question $x_i \in \mathcal{X}$: $\{y_i^k\}_{k=1}^K \sim \pi_0(\cdot|x_i)$
5:     Label binary reward for each solution $y_i^k$: $r_i^k = r^{\text{sl}}(y_i^k, y_i^*)$
6:     Append to rejection sampling set: $\mathcal{D}_{\text{rjs}} \leftarrow \mathcal{D}_{\text{rjs}} \cup \{(x_i, y_i^k, r_i^k)\}$
7:     // Obtain verifications
8:     Choose the best/worst-of-N samples: $k^+ = \arg\max_k r_i^k, k^- = \arg\min_k r_i^k$
9:     **if** $r_i^{k^+} = 0$ or $r_i^{k^-} = 1$ **then**
10:       continue // All correct or all incorrect solutions
11:     **else**
12:       $y_i^* \leftarrow y_i^{k^+}, \text{c\_flag} \leftarrow \textbf{false}, \text{i\_flag} \leftarrow \textbf{false}$
13:       **for** $k = 1, \ldots, K$ **do**
14:         **if** $r_i^k = 0$ **then**
15:           $v_i^- \sim \pi_0(\cdot|x_i, y_i^k, y_i^*)$
16:           **if** $f^{\text{vf}}(v_i^-) = 1$ **then**
17:             $\text{i\_flag} \leftarrow 1$
18:             $\mathcal{D}_{\text{pair}} \leftarrow \mathcal{D}_{\text{pair}} \cup \{(x_i, y_i^k, v_i^-, y_i^*)\}$
19:           **end if**
20:         **else if** $r_i^k = 1$ and $k \neq k^+$ **then**
21:           $v_i^+ \sim \pi_0(\cdot|x_i, y_i^k, y_i^*)$
22:           **if** $f^{\text{vf}}(v_i^+) = 1$ **then**
23:             $\text{c\_flag} \leftarrow 1$
24:             $\mathcal{D}_{\text{pair}} \leftarrow \mathcal{D}_{\text{pair}} \cup \{(x_i, y_i^k, v_i^+)\}$
25:           **end if**
26:         **end if**
27:         **if** $\text{c\_flag} = 1$ and $\text{i\_flag} = 1$ **then**
28:           break
29:         **end if**
30:       **end for**
31:     **end if**
32: **end for**
33: **return** $\mathcal{D}_{\text{pair}}$

---

---

**Algorithm 3** RAFT Message-wise Policy Optimization

---

1: **Inputs:** Question-answer batch $\mathcal{D}_i = \mathcal{X}_i \times \mathcal{Y}_i = \{(x_j, y_j^*)\}_{j=1}^B$, batch size $B$, policy model $\pi_\theta$, number of rollouts per question $K$, generated trajectory $\{\tau_j^k\}_{k=1}^K$, solution correctness rewards $\{r_{j,k,l}^{\text{sl}}\}_{k\in[K],l\in[L_k]}$, verification correctness rewards $\{r_{j,k,l}^{\text{vf}}\}_{k\in[K],l\in[L_k]}$

2: Choose the best-of-N trajectory for each question $x_j$ based on last solution message:
$$k^+ = \arg\max_k r_{j,k,L_k}^{\text{sl}}$$

3: // Apply constraint

4: Filter out questions with no correct final solution or no correct verification, i.e. learning batch is
$$\mathcal{D}_{\text{learn}} = \left\{ x_j, \tau_j^{k^+}, \{r_{j,k^+,l}^{\text{sl}}\}_{l\in[L_k]}, \{r_{j,k^+,l}^{\text{vf}}\}_{l\in[L_k]} \mid r_{j,k^+,L_k}^{\text{sl}} = 1 \vee r_{j,k^+,l}^{\text{vf}} = 1 \right\}_{j\in[B]}$$

5: Perform one gradient update on $\theta$ with Equations (2) and (3) using $\mathcal{D}_{\text{learn}}$

---

**Algorithm 4** RLOO Message-wise Policy Optimization

---

1: **Inputs:** Question-answer batch $\mathcal{D}_i = \mathcal{X}_i \times \mathcal{Y}_i = \{(x_j, y_j^*)\}_{j=1}^B$, batch size $B$, policy model $\pi_\theta$, number of rollouts per question $K$, generated trajectory $\{\tau_j^k\}_{k=1}^K$, solution correctness rewards $\{r_{j,k,l}^{\text{sl}}\}_{k\in[K],l\in[L_k]}$, verification correctness rewards $\{r_{j,k,l}^{\text{vf}}\}_{k\in[K],l\in[L_k]}$

2: // Message-wise advantage

3: **for** $l = 1, \ldots, \max_k L_k; r = r^{\text{sl}}, r^{\text{vf}}$ **do**

4: $\quad \mu_{j,l} = \frac{1}{K} \sum_{k\in[K]} r_{j,k,l}$

5: $\quad \sigma_{j,l} = \left( \frac{1}{K} \sum_{k\in[K]} |r_{j,k,l} - \mu_{j,l}|^2 \right)^{\frac{1}{2}}$

6: $\quad A_{j,k,l} = \frac{r_{j,k,l} - \mu_{j,l}}{\sigma_{j,l}}$

7: **end for**

8: Learning batch contains all $K$ samples for each question:
$$\mathcal{D}_{\text{learn}} = \left\{ x_j, \tau_j^k, \{A_{j,k,l}^{\text{sl}}\}_{l\in[L_k]}, \{A_{j,k,l}^{\text{vf}}\}_{l\in[L_k]} \right\}_{j\in[B],k\in[K]}$$

9: Perform one gradient update on $\theta$ with Equations (2) and (3) using $\mathcal{D}_{\text{learn}}$

---

## B  Experimental setup details

**Tasks.**  We evaluate model on test sets as follows:

- MATH500 (Lightman et al., 2023). A dataset of 500 problems selected from the full MATH (Hendrycks et al., 2021) evaluation set. This test set spans five difficulty levels and seven subjects, which promotes a comprehensive evaluation of reasoning capabilities.

- AMC23. A dataset of 40 problems from the American Mathematics Contest 12 (AMC12) 2023 (AI-MO, 2023). This test set consists of challenging competition questions intending to evaluate the model's capability to solve complex reasoning problems.

- AIME24. A dataset of 30 problems from the American Invitational Mathematics Examination (AIME) 2024 (AI-MO, 2024). This test set contains difficult questions, with few at AMC level and others drastically more difficult in comparison, aim to access the model's abiblity to perform more intricate math reasoning.

**Implementation details.**  We use the AdamW optimizer with $\beta_1 = 0.9$, $\beta_2 = 0.95$, weight decay $= 0.1$, and a constant learning rate $1.0 \times 10^{-6}$. We conduct all training runs on 32 NVIDIA H100 GPUs. We set the global batch size to 2048, and train for 256 steps.

## C  Extra results

### C.1  Verifier reliability

We provide detailed diagnostics for verifier reliability in Table 5. Each confusion matrix corresponds to a base model and task pair, with the rows and columns indicating the actual and predicted solution correctness, respectively - i.e., diagonal cells represent the true positive (TP) and true negative (TN) rates while the off-diagonal cells represent the false positive (FP) and false negative (FN) rates. We observe the following phenomena:

- On easier tasks, the proposer has higher solution accuracy, and the verifier tends to show higher TP&FP and lower TN&FN.

- Stronger models that reach higher solution accuracy also have higher TP&FP.

- The small model's high verification accuracy attributes largely to its higher TN.

| Base Model | MATH500 | | AMC2023 | | AIME2024 | |
|---|---|---|---|---|---|---|
| Llama-3.1-8B-Instruct | 90.2 (266/295) | 9.8 (29/295) | 81.9 (9/11) | 18.2 (2/11) | 0 (0/1) | 100 (1/1) |
| | 34.1 (70/205) | 65.9 (135/205) | 24.1 (7/29) | 75.9 (22/29) | 0 (0/29) | 100 (29/29) |
| Llama-3.1-70B-Instruct | 100 (385/385) | 0 (0/385) | 100 (21/21) | 0 (0/21) | 85.7 (6/7) | 14.3 (1/7) |
| | 87.0 (100/115) | 13.0 (15/115) | 84.2 (16/19) | 15.8 (3/19) | 82.6 (19/23) | 17.4 (4/23) |
| Llama-3.3-70B-Instruct | 99.0 (385/389) | 1.0 (4/389) | 93.1 (27/29) | 6.9 (2/29) | 100 (7/7) | 0 (0/7) |
| | 78.4 (87/111) | 21.6 (24/111) | 72.7 (8/11) | 27.3 (3/11) | 82.6 (19/23) | 17.4 (4/23) |

Table 5: Diagnostics for verifier reliability at the first turn across MATH500, AMC2023, and AIME2024 benchmarks.

### C.2  Per-turn performance analysis

We provide the per-turn performance statistics for AIME24 and AMC23 in Table 6 and Table 7, respectively. The results are consistent with MATH500 analysis in Table 2. SPOC generally improves or maintains performance on the second solution turns. The smaller model has lower final accuracy yet larger turn-wise improvements, while larger models tend to achieve correct solutions sooner at turn1. Moreover, turn-wise corrections occurs less in these two challenging competition benchmarks, as they contain significantly fewer

| Base Model trained w/ SPOC | Base.Acc. | Verif.Acc.@t1 | Acc.@t1 | Acc.@t2 | $\Delta(t1, t2)$ | $\Delta_{c \to i}$ | $\Delta_{i \to c}$ |
|---|---|---|---|---|---|---|---|
| Llama-3.1-8B-Instruct | 3.3 | 29/30 | 1/30 | 2/30 | 1/30 | 0/1 | 1/7 |
| Llama-3.1-70B-Instruct | 16.7 | 10/30 | 7/30 | 7/30 | 0/30 | 0/1 | 0/1 |
| Llama-3.3-70B-Instruct | 26.7 | 11/30 | 7/30 | 7/30 | 0/30 | 0 | 0/1 |

Table 6: Performance across first two solution turns on AIME2024. $\Delta_{c \to i}$ and $\Delta_{i \to c}$ indicate $\frac{n_{c \to i}}{n_{c \to c} + n_{c \to i}}$ and $\frac{n_{i \to c}}{n_{i \to i} + n_{i \to c}}$ over turns, respectively.

| Base Model trained w/ SPOC | Base.Acc. | Verif.Acc.@t1 | Acc.@t1 | Acc.@t2 | $\Delta(t1, t2)$ | $\Delta_{c \to i}$ | $\Delta_{i \to c}$ |
|---|---|---|---|---|---|---|---|
| Llama-3.1-8B-Instruct | 22.5 | 31/40 | 27.5 | 32.5 | 5.0 | 0/2 | 2/11 |
| Llama-3.1-70B-Instruct | 32.5 | 24/40 | 21/40 | 21/40 | 0 | 0 | 0 |
| Llama-3.3-70B-Instruct | 57.5 | 30/40 | 29/40 | 28/40 | -2.5 | 1/2 | 0/2 |

Table 7: Performance across first two solution turns on AMC2023. $\Delta_{c \to i}$ and $\Delta_{i \to c}$ indicate $\frac{n_{c \to i}}{n_{c \to c} + n_{c \to i}}$ and $\frac{n_{i \to c}}{n_{i \to i} + n_{i \to c}}$ over turns, respectively.

questions than MATH500. We will include both tables in the appendix of our revised manuscript.

Table 2 presents our per-turn performance analysis over turn1 → 2, where the majority of self-correction occurs. In practice, all finetuned models perform multiple rounds of self-reflection. We hereby present the complete results, where the Table 8 shows the turn 2 → 3 performance of all models, and Table 9 shows the all-turn performance of the 8B model (as the other stopped reflection earlier). Results suggest that the 8B model reaches a maximum of 6 turns while the 70B models reach a maximum of 3 turns across all 500 evaluation questions. This observation aligns with our discussion in Section 4.2, where stronger models tend to achieve correct solutions sooner. We also observe that the amount of questions requiring additional solutions drops over turns, aligning with the looping until verified correctness behavior. Overall, SPOC achieves improvement over turns.

| Base Model trained w/ SPOC | Base.Acc. | Verif.Acc.@t2 | Acc.@t2 | Acc.@t3 | $\Delta(t2, t3)$ | $\Delta_{c \to i}$ | $\Delta_{i \to c}$ |
|---|---|---|---|---|---|---|---|
| Llama-3.1-8B-Instruct | 52.2 | 19/22 | 61.0 | 61.2 | 0.2 | 0/3 | 1/18 |
| Llama-3.1-70B-Instruct | 65.8 | 0 | 77.4 | 77.4 | 0 | - | - |
| Llama-3.3-70B-Instruct | 75.6 | 4/24 | 77.8 | 77.8 | 0 | - | - |

Table 8: Performance across solution turns 2 → 3 on MATH500. $\Delta_{c \to i}$ and $\Delta_{i \to c}$ indicate $\frac{n_{c \to i}}{n_{c \to c} + n_{c \to i}}$ and $\frac{n_{i \to c}}{n_{i \to i} + n_{i \to c}}$ over turns, respectively.

| Turn $l$ | Verif.Acc.@$t_l$ | Acc.@$t_l$ | Acc.@$t_{l+1}$ | $\Delta(t_l, t_{l+1})$ | $\Delta_{c \to i}$ | $\Delta_{i \to c}$ |
|---|---|---|---|---|---|---|
| 1 | 401/500 | 59.0 | 61.0 | 2.0 | 8/29 | 18/79 |
| 2 | 19/22 | 61.0 | 61.2 | 0.2 | 0/3 | 1/18 |
| 3 | 6/8 | 61.2 | 61.0 | -0.2 | 2/2 | 1/6 |
| 4 | 2/2 | 61.0 | 61.0 | 0.0 | - | 0/2 |
| 5 | 1/1 | 61.0 | 61.0 | 0.0 | - | 0/1 |
| 6 | 0/1 | 61.0 | - | - | - | - |

Table 9: Performance across all solution turns on MATH500 for Llama-3.1-8B-Instruct base model. $\Delta_{c \to i}$ and $\Delta_{i \to c}$ indicate $\frac{n_{c \to i}}{n_{c \to c} + n_{c \to i}}$ and $\frac{n_{i \to c}}{n_{i \to i} + n_{i \to c}}$ over turns, respectively.

## D Preliminaries

CGPO (Xu et al., 2024) is a constrained RL framework that allows for flexible applications of constraints on model generations. Denoting the constraints that the LLM generations need to satisfy as $\{C_1, \ldots, C_M\}$, the prompt-generation set that satisfies constraint $C_m$ is defined as $\Sigma_m = \{(x, y) \in \mathcal{X} \times \mathcal{Y} : (x, y) \text{ satisfies } C_m\}$. The feasible region is defined as

the prompt-generation set that satisfies all constraints, i.e., $\Sigma = \cap_{m=1}^{M} C_m$. In the single-task setting, CGPO solves the constrained optimization problem as follows:

$$\max_{\theta} \quad \mathbb{E}_{x \sim \mathcal{X}, y \sim \pi_\theta(x)} \left[ r(x, y) \right]$$

$$\text{s.t.} \quad \mathbb{P}_{x \sim \mathcal{X}, y \sim \pi_\theta(x)} \left( (x, y) \in \Sigma \right) > 0,$$

$$\text{KL}_{x \sim \mathcal{X}} \left( \pi_\theta(x) \| \pi_{\text{ref}}(x) \right) \leq \text{KL}_{\max}$$

where $r(x, y)$ is the reward function. CGPO is compatible with a wide spectrum of policy optimizers. The RAFT (Dong et al., 2023) algorithm prompts the current policy to generate multiple responses for each prompt, and the best-of-N (BoN) response is used to perform a one-step SFT update on the policy.

# E   Prompts

```
Llama 3.1 COT query template

User:
Solve the following math problem efficiently and clearly:

- For simple problems (2 steps or fewer):
Provide a concise solution with minimal explanation.

- For complex problems (3 steps or more):
Use this step-by-step format:

## Step 1: [Concise description]
[Brief explanation and calculations]

## Step 2: [Concise description]
[Brief explanation and calculations]

...

Regardless of the approach, always conclude with:

Therefore, the final answer is: $\\boxed{answer}$. I hope it is correct.

Where [answer] is just the final number or expression that solves the problem.

Problem: {{ Question }}
```

Figure 4: Llama 3.1 COT query template (Dubey et al., 2024).

```
Simple COT query template

User:
Please reason step by step, and put your final answer within \\boxed{}.
Question: {{ Question }}
```

Figure 5: Simple COT query template (Guo et al., 2025).

---

**Instance reflection query template**

**User:**

Compare a correct solution and a potential solution to the same math question, identify the error in the potential solution, if any, and briefly explain the error in two sentences or less in the <reflection> ... </reflection> tags, and then conclude with:

Verification: Is the previous solution correct? (Yes/No)

The explanation should be "I think the solution is correct." if the potential solution is correct.

DO NOT solve the question. DO NOT include phrases implying the comparison between the two solutions.

Question: {{ Question }}

Correct solution: {{ correct_solution }}

Potential solution: {{ potential_solution }}

---

Figure 6: Instance reflection query template.

---

**SPOC simple COT query template**

**User:**

Please reason step by step, and put your final answer within \\boxed{}.

After each solution attempt, reflect on its correctness within <reflection> ... </reflection> tags.

Your reflection should first concisely evaluate the previous solution, and then conclude with:

Verification: Is the previous solution correct? (Yes/No)

If the verification is "No", rewrite the solution in a separate attempt, either correcting the error or choosing a different approach altogether.

Question: {{ Question }}

---

Figure 7: SPOC simple COT query template.

---

**Self-Refine w/o oracle query template**

**User:**

There might be an error in the solution above because of lack of understanding
of the question.
Please correct the error, if any, and rewrite the solution.
Be sure to apply the given format and conclude with:
"Therefore, the final answer is: $\\boxed{answer}$."

---

Figure 8: Self-Refine w/o oracle query template (Madaan et al., 2023).

---

**Self-Refine w/ oracle query template**

**User:**

There is an error in the solution above because of lack of understanding of
the question.
Please correct the error and rewrite the solution.
Ensure you use the information from past attempts.
If you arrive at a solution you have already had, the answer is incorrect once
again, so take that into account and retry if necessary.
Be sure to apply the given format and conclude with:
"Therefore, the final answer is: $\\boxed{answer}$."

---

Figure 9: Self-Refine w/ oracle query template (Madaan et al., 2023).

## F   Example response

We present example responses of SPOC finetuned on Llama-3.1-70B-Instruct on MATH500.

---

**SPOC sample response**

*Problem info:*

Problem: Positive integers $a$, $b$, and $2009$, with $a<b<2009$, form a geometric
sequence with an integer ratio. What is $a$?
Answer: 41
Subject: Algebra
Level: 4
Unique_id: test/algebra/791.json

**User:**

Please reason step by step, and put your final answer within \\boxed{}.

After each solution attempt, reflect on its correctness within <reflection> ...
</reflection> tags.

Your reflection should first concisely evaluate the previous solution, and then
conclude with:

Verification: Is the previous solution correct? (Yes/No)

If the verification is "No", rewrite the solution in a separate attempt, either
correcting the error or choosing a different approach altogether.

Question: Positive integers $a$, $b$, and $2009$, with $a<b<2009$, form a geometric
sequence with an integer ratio. What is $a$?

---

**Assistant:**

## Step 1: Understand the problem and recall the formula for a geometric sequence.
A geometric sequence is defined by the formula $a, ar, ar^2, \ldots, ar^{n-1}$, where $a$ is the first term and $r$ is the common ratio. In this problem, we have $a, b, 2009$ as the terms of the geometric sequence, with $a < b < 2009$. Therefore, we can write $b = ar$ and $2009 = ar^2$.

## Step 2: Express $b$ in terms of $a$ and $r$, and $2009$ in terms of $a$ and $r$ to find the relationship between $a$ and $r$.
Given $b = ar$ and $2009 = ar^2$, we can see that $r = \frac{b}{a}$ and $r^2 = \frac{2009}{a}$. Since $r$ is an integer, $r^2$ must also be an integer, which means $\frac{2009}{a}$ must be an integer. Therefore, $a$ must be a factor of 2009.

## Step 3: Find the factors of 2009 to determine possible values for $a$.
To find the factors of 2009, we can start by checking for its prime factorization. 2009 is a prime number, which means its only factors are 1 and 2009. However, since $a < b < 2009$, $a$ cannot be 2009. Thus, the only possible value for $a$ is 1, but we need to verify if this satisfies the condition of forming a geometric sequence with an integer ratio.

## Step 4: Verify if $a = 1$ satisfies the conditions of the problem.
If $a = 1$, then $b = ar = 1 \cdot r = r$ and $2009 = ar^2 = 1 \cdot r^2 = r^2$. This implies $r^2 = 2009$, but since 2009 is not a perfect square (it is a prime number), $r$ cannot be an integer. Therefore, $a = 1$ does not satisfy the condition of having an integer ratio $r$. We need to reconsider our approach based on the prime factorization of 2009 and the properties of geometric sequences.

## Step 5: Reconsider the prime factorization of 2009 and its implications for $a$ and $r$.
Given that 2009 is a prime number, for $a, b, 2009$ to form a geometric sequence with $a < b < 2009$ and an integer ratio $r$, we must have $b = ar$ and $2009 = ar^2$. Since $r$ is an integer and $2009 = ar^2$, $r^2$ must divide 2009. However, since 2009 is prime, the only integer divisors of 2009 are 1 and 2009 itself. This means $r^2 = 2009$ is not possible for any integer $r$ since 2009 is not a perfect square. The only other option is $r^2 = 1$, which gives $r = 1$ or $r = -1$, but $r = 1$ would not give a geometric sequence with distinct terms, and $r = -1$ would not satisfy $a < b < 2009$ since multiplying by -1 would change the order of the terms. We need to think about the factors of 2009 in a different way, considering that $a$ and $r$ must both be integers and $a < b < 2009$.

## Step 6: Consider the factorization of 2009 in the context of geometric sequences.
Since 2009 is a prime number, for $a, b, 2009$ to be in a geometric sequence, we have to think about how $a$ and $r$ can be chosen such that $b = ar$ and $2009 = ar^2$ are both integers and $a < b < 2009$. Given $2009 = ar^2$, and knowing 2009 is prime, we realize that for $r^2$ to divide 2009, $r^2$ must be 1 (since 2009 is prime and has no other divisors besides 1 and itself). This implies $r = 1$ or $r = -1$, but as discussed, $r = 1$ does not yield a geometric sequence with distinct terms, and $r = -1$ does not satisfy the condition $a < b < 2009$ because multiplying by -1 would reverse the order. We must reconsider the relationship between $a$, $b$, and 2009 in terms of their prime factorization and the properties of geometric sequences.

## Step 7: Apply the correct relationship between $a$, $b$, and 2009 considering the geometric sequence and integer ratio.
Given $b = ar$ and $2009 = ar^2$, dividing the second equation by the first gives $\frac{2009}{b} = r$. Since $r$ is an integer, $b$ must be a factor of 2009.
The factors of 2009 are 1 and 2009 (since 2009 is prime), but $b$ cannot be 2009 because $b < 2009$. Thus, $b = 1$ is not possible since $a < b$, implying $a$ would have to be less than 1, which is not possible for positive integers. We

need to find a factor of 2009 that fits the geometric sequence criteria. However, our analysis has been misguided because we overlooked the simplest factorization that allows for a geometric sequence with integer ratio. Let's correct this by directly applying the condition that $a, b, 2009$ form a geometric sequence with $r$ being an integer.

## Step 8: Correctly apply the geometric sequence formula and the fact that 2009 is a prime number.
Since $2009 = ar^2$ and $r$ is an integer, we look for integer factors of 2009. However, since 2009 is prime, its only factors are 1 and 2009. For $a, b, 2009$ to be in a geometric sequence, we must have $b = ar$ and $2009 = ar^2$. Given $r$ is an integer and $a < b < 2009$, we examine the possibility of $r = 2009$ and $a = 1$, but this does not fit because $b$ would then be 2009, violating $b < 2009$. The correct approach involves recognizing that since 2009 is prime, the only way $a, b, 2009$ can form a geometric sequence with integer $r$ is if $r$ itself is a factor of 2009, but since 2009 is prime, $r$ must effectively be 1 or 2009 for the sequence to hold with integer values. However, $r = 1$ does not yield a geometric sequence with distinct terms, and $r = 2009$ does not fit the $a < b < 2009$ condition. We need to think about the factors of 2009 in a way that allows for a geometric sequence with $a < b < 2009$ and $r$ being an integer.

## Step 9: Re-evaluate the conditions for $a$, $b$, and 2009 to form a geometric sequence.
Given $b = ar$ and $2009 = ar^2$, and knowing 2009 is prime, for $r$ to be an integer, $a$ must be such that $ar^2 = 2009$ holds true with $r$ being an integer. Since 2009 is prime, $r^2$ must divide 2009, implying $r^2 = 1$ or $r^2 = 2009$ since these are the only divisors of 2009. $r^2 = 2009$ is not possible for integer $r$ since 2009 is not a perfect square. Thus, $r^2 = 1$, giving $r = 1$ or $r = -1$. However, $r = 1$ does not yield a geometric sequence with distinct terms, and $r = -1$ would imply $b = -a$, which does not fit $a < b < 2009$ for positive integers. We must consider the nature of geometric sequences and prime numbers to find $a$.

## Step 10: Apply the properties of geometric sequences and prime numbers correctly.
For $a, b, 2009$ to be a geometric sequence with integer $r$, and given 2009 is prime, we look for $a$ and $r$ such that $2009 = ar^2$ and $b = ar$. Since $r$ must be an integer and 2009 is prime, $r^2$ must be a factor of 2009, which means $r^2 = 1$ (since 2009 is prime and has no other square factors). This implies $r = 1$ or $r = -1$, but $r = 1$ does not give a geometric sequence with distinct terms, and $r = -1$ does not fit the condition $a < b < 2009$ for positive $a$ and $b$. We need to think about the prime factorization of 2009 and how it applies to geometric sequences. Given 2009 is prime, for $a, b, 2009$ to form a geometric sequence, $a$ must be 1, and $r$ must be such that $r^2 = 2009$, but since 2009 is not a perfect square, this approach is flawed. We must reconsider the basic properties of geometric sequences and the given conditions.

## Step 11: Correctly identify $a$ based on the geometric sequence and prime number conditions.
Given $a, b, 2009$ form a geometric sequence with integer $r$, and 2009 is prime, we have $b = ar$ and $2009 = ar^2$. Since $r$ is an integer and 2009 is prime, $r^2$ must divide 2009, implying $r^2 = 1$ or $r^2 = 2009$. Since $r^2 = 2009$ is not possible for integer $r$ (2009 is not a perfect square), we consider $r^2 = 1$, giving $r = 1$ or $r = -1$. However, $r = 1$ does not yield a geometric sequence with distinct terms, and $r = -1$ does not fit $a < b < 2009$ for positive integers. The error lies in assuming $r^2$ must directly divide 2009; instead, we should consider how $a$ and $r$ relate through the geometric sequence formula. Since 2009 is prime, and $a, b, 2009$ are in a geometric sequence, we look for $a$ and $r$ such that $2009 = ar^2$ and $b = ar$, with $r$ being an integer. Given 2009's primality, $a$ must be 1 for $r^2$ to be an integer factor of 2009, but since 2009 is not a perfect square, $r$ cannot be an integer if $a = 1$. We need to reconsider the relationship between $a$, $b$, and 2009 in the context of geometric sequences and prime numbers.

## Step 12: Finalize the correct approach for finding $a$.
Given $a, b, 2009$ form a geometric sequence with $r$ as an integer, and 2009 is prime, we must find $a$ such that $b = ar$ and $2009 = ar^2$. Since 2009 is prime, for $r^2$ to divide 2009, $r^2$ must be 1 (since 2009 is not a perfect square). This implies $r = 1$ or $r = -1$, but $r = 1$ does not yield a geometric sequence with distinct terms, and $r = -1$ does not fit $a < b < 2009$ for positive $a$ and $b$. The key insight is recognizing that since 2009 is prime, $a$ must be such that $ar^2 = 2009$ holds with $r$ being an integer. Given $r^2 = 1$ is the only integer square factor of 2009 (since 2009 is not a perfect square), we must consider how $a$ and $r$ relate through the geometric sequence. If $r = -1$, then $b = -a$, which does not fit $a < b < 2009$ for positive integers. Thus, we need to think about the factors of 2009 in a way that allows for a geometric sequence. Since 2009 is prime, the only factors are 1 and 2009, implying $a$ could be 1 if $r^2 = 2009$, but $r$ cannot be an integer in this case. The correct approach involves recognizing that $a$ must be a factor of 2009 such that $r$ can be an integer, but since 2009 is prime, $a$ must effectively be 1 for $r$ to be an integer, given $r^2 = 1$ as the only viable integer square factor of 2009.

Therefore, the final answer is: \boxed{1}<|eom_id|>

<reflection>
The error lies in the incorrect assumption that $r^2$ must directly divide 2009, leading to a flawed analysis of possible values for $r$ and $a$. This mistake results in an incorrect conclusion that $a$ must be 1, when in fact, $a$ should be derived from considering the factors of 2009 and the properties of geometric sequences with integer ratios.

Verification: Is the previous solution correct? No
</reflection><|eom_id|>

## Step 1: Understand the problem and recall the formula for a geometric sequence.
A geometric sequence is defined by the formula $a_n = a_1 \cdot r^{(n-1)}$, where $a_n$ is the nth term of the sequence, $a_1$ is the first term, $r$ is the common ratio, and $n$ is the term number. For the given problem, we have $a_1 = a$, $a_2 = b$, and $a_3 = 2009$. Therefore, we can write the equations $b = ar$ and $2009 = ar^2$.

## Step 2: Express $b$ in terms of $a$ and $r$, and $2009$ in terms of $a$ and $r$.
From the geometric sequence formula, we have $b = ar$ and $2009 = ar^2$.

## Step 3: Solve for $r$ using the information given.
Dividing the equation $2009 = ar^2$ by $b = ar$, we get $\frac{2009}{b} = r$. Since $b = ar$, we can substitute $b$ in the equation to get $\frac{2009}{ar} = r$, which simplifies to $2009 = ar^2$. We already know this, so let's directly use the relationship between $a$, $b$, and $2009$ to find $r$. Given $b = ar$ and $2009 = ar^2$, dividing these equations gives $r = \frac{2009}{b}$.

## Step 4: Determine the possible values of $r$ and $a$.
Since $a$, $b$, and $2009$ form a geometric sequence with $a < b < 2009$, and $r$ is an integer, we need to find integer factors of $2009$ that could serve as $r$. The prime factorization of $2009$ is $7^2 \cdot 41$. Thus, possible values for $r$ could be $1$, $7$, $41$, or $49$, considering the factors of $2009$. However, since $a < b < 2009$, $r$ must be greater than $1$.

## Step 5: Test possible values of $r$ to find $a$.
Let's test $r = 7$: If $r = 7$, then $2009 = a \cdot 7^2$, which gives $a = \frac{2009}{49}$. Calculating this, $a = 41$, which is an integer and satisfies $a < b < 2009$.
Let's test $r = 41$: If $r = 41$, then $2009 = a \cdot 41^2$, which does not yield an integer $a$ since $41^2 = 1681$ and $2009 / 1681$ is not an integer.
Let's test $r = 49$: If $r = 49$, then $2009 = a \cdot 49^2$ does not apply since

```
\(49^2 = 2401\), which is larger than \(2009\), making it impossible for \(a\) to be an
integer in this context.

## Step 6: Verify the solution.
Given \(r = 7\) and \(a = 41\), we find \(b = ar = 41 \cdot 7 = 287\). Thus, the sequence
is \(41, 287, 2009\), which satisfies the given conditions.

The final answer is: $\boxed{41}$<|eom_id|>

<reflection>
I think the solution is correct.\n\nVerification: Is the previous solution correct? Yes
</reflection><|eot_id|>
```

