# OpenReview forum: "Boosting LLM Reasoning via Spontaneous Self-Correction"
_colmweb.org/COLM/2025/Conference — COLM 2025_

### Official Review · Reviewer_VSGv · 2025-05-10

**Rating:** 7
**Confidence:** 3
**Ethics Flag:** 1

**Summary:**

The paper argues that the existing methods rely on external system prompts for verification and solution generation to perform self-correction, which may be cumbersome to deploy. To this end, the paper proposes SPOC, a technique to allow the model to perform self-correction adaptively by acting as a solution generator and verifier in a single inference call. Specifically, the model is first trained with paired SFT data which motivates the model to perform multi-turn generation-verification. Subsequently, the SPOC performs RL-tuning to explore diverse scenarios on the fly and learn from the oracle rewards (e.g., solution correctness and Yes/No correctness for verifier). Interestingly, the model policy is optimized for both solution generation and verification jointly. The paper achieves large gains over the baseline models across diverse math datasets.

**Questions To Authors:**

- In Line 38, the authors mention that lack of adaptive self-reflection leads to “ineffective test-time scaling”. This statement does not make a lot of sense since prior work has studied test-time scaling in traditional self-reflection too [1].
- In Table 1, the LLama-8B-Instruct Self-Refine w/ Oracle achieves better performance than SPOC on AMC-23 but it is not highlighted. Feel free to fix the typo.
- I think the paper presentation requires a lot more work. For instance, Figure 3 presented in Page 4 is not relevant uptill the experiment in page 9 which is quite confusing.

[1] https://arxiv.org/abs/2501.19306

**Reasons To Accept:**

- The paper highlights an important problem that closed loop self-verification/correction can be cumbersome at inference due to heavy reliance on different prompting or allocation of multiple models to act as solution/verification generators.
- The proposed solution of performing self-correction in the single inference call is interesting and allows the model to make decisions adaptively.
- The performance improvements over the relevant baselines is quite strong across different benchmarks.

**Reasons To Reject:**

- While the presentation and theory in the paper is general-purpose, the experiments were performed with a lesser number of self-correction turns. For instance, the D_pair (Line 186) data indicates that the model is taught to self-correct within one-round (question, wrong answer, verification, right answer). In reality, you might need multiple rounds of self-correction. Table 2 also suggests that there is only one-round of self-reflection in the experiments. However, Figure 2 (Right) indicates that there are multiple rounds of self-correction where an example goes through three rounds of solution generation till the final verification is correct. It is confusing how the model is generating more rounds in the RL stage than it has been taught in the SFT stage.
- It is unclear how the method is required in the light of thinking models that can perform self-verification/self-correction before generating the final answer. The introduction never touches on this aspect. We observe that the authors show that SPOC works on top of Deepseek-Distill models too so it will be helpful to have more nuanced discussion on why self-correction is needed for thinking models too. Related to the above, it is not clear how well the method performs beyond one-round of self-reflection. In contrast, the thinking models can perform multiple rounds of self-verification, backtracking, alternative solutions, and reflection in their reasoning traces.

[1] https://arxiv.org/abs/2501.12948

---

> ### Author Response · Authors · 2025-06-03
> **Response to Reviewer VSGv (1/2)**
>
> We appreciate your constructive feedback and the time you have devoted. Please kindly find our detailed responses to your concerns as follows.
>
> > **Q1 Self-correction turns.** While the presentation and theory in the paper is general-purpose, the experiments were performed with a lesser number of self-correction turns. ... It is confusing how the model is generating more rounds in the RL stage than it has been taught in the SFT stage.
>
> **A1** We firstly would like to clarify that D_pair contains both one-turn (question, correct solution, verification) and two-turn (question, incorrect solution, verification, correct solution) trajectories, as described in Section 3.2 and Algorithm 2. We will include both cases in Figure 2 (left) accordingly.
>
> Besides, we would like to confirm that PairSFT with one-turn *(without correction)* and two-turn *(with correction)* generations successfully initiates the dynamic multi-turn behavior. Our formulation and training data promotes spontaneous message and adaptive generation based on verification outcomes (see prompt in appendix for reference). Therefore, the looping until verified correctness behavior in Figure 2 (right) is an accurate illustration of the rollouts in practice.
>
> Table 2 presents our per-turn performance analysis over turn1->2, where the majority of self-correction occurs. In practice, all finetuned models perform multiple rounds of self-reflection. We hereby present the complete results below, where the first table shows the turn2->3 performance of all models, and the second table shows the all-turn performance of the 8B model (as the other stopped reflection earlier). Results suggest that the 8B model reaches a maximum of 6 turns while the 70B models reach a maximum of 3 turns across all 500 evaluation questions. This observation aligns with our discussion in Section 4.2, where stronger models tend to achieve correct solutions sooner. We also observe that the amount of questions requiring additional solutions drops over turns, aligning with the looping until verified correctness behavior. Overall, SPOC achieves improvement over turns.
>
> We will include both tables in the appendix of our revised manuscript.
>
> > **Q2 Discussions on thinking models.** It is unclear how the method is required in the light of thinking models that can perform self-verification/self-correction before generating the final answer. ... We observe that the authors show that SPOC works on top of Deepseek-Distill models too so it will be helpful to have more nuanced discussion on why self-correction is needed for thinking models too.
>
> **A2** SPOC triggers self-verification/correction based on its judged correctness of the previous message, which introduces binary correctness labels for each message. Hence, SPOC applies process supervision to reward correct and penalize incorrect *solutions and verifications*. Our ablations underscore the importance of such process-level supervision. Relying solely on final answer correctness, thinking model’s outcome reward does not supervise intermediate reflection steps within the long CoT, potentially resulting in undesirable excessively long reasoning [2,3] and hindering efficient compute usage.
>
> SPOC is compatible with R1’s long thinking, demonstrated by experiments on DeepSeek-R1-Distill models. These experiments not only support our accessesment of SPOC’s effectiveness and generalizability across initial model capabilities, but also serve as an exploration towards reflective reasoning in long CoT. As noted in Section 5, to address the prohibitive length of long CoTs, an interesting future direction is to extend SPOC to partial solutions in long CoTs, using step-level process rewards to guide RL training and enable dynamic revisions when errors are detected until reaching the final answer.
>
> > **Q3 Related work.** In Line 38, ... prior work has studied test-time scaling in traditional self-reflection too.
>
> **A3** We appreciate the reviewer for bringing up [4]. This concurrent work is akin to SPOC in a sense that at inference time self-correction is triggered by results of self-verification, which differs from traditional reflection works [1,5]. Differently, [4] involves no training to initiate/improve spontaneous reflection, thereby requiring an explicit prompting to generate each single message, while SPOC performs real-time adaptive reflective reasoning in a single inference pass, allows for more flexible deployment. We will discuss [4] in related works and revise our presentation in our next version of the manuscript.
>
> *(continued next)*

---

> > ### Author Response · Authors · 2025-06-03
> > **Response to Reviewer VSGv (2/2)**
> >
> > *(continuing)*
> >
> > > **Q4 Typo.** In Table 1, the LLama-8B-Instruct Self-Refine w/ Oracle achieves better performance than SPOC on AMC-23 but it is not highlighted. Feel free to fix the typo.
> >
> > **A4** When marking the best performance, we omitted the prompting-based Self-Refine w/ Oracle for fair comparisons as it relies on oracle correctness labels unavailable to other approaches. We will use a different symbol to mark it in our next revision of the manuscript.
> >
> > > **Q5 Presentation.** I think the paper presentation requires a lot more work. For instance, Figure 3 presented in Page 4 is not relevant uptill the experiment in page 9 which is quite confusing.
> >
> > **A5** We are grateful for your constructive comments. Figure 3 is intended to illustrate the reward setting discussed on page 5 and ablation variants on page 9. We will improve our organization and overall presentation in our next revision.
> >
> > ---
> >
> > **Per-turn performance analysis: turn2->3**
> > | Base model | base.acc. | verif.acc.@$t_{2}$ | acc.@$t_{2}$ | acc.@$t_{3}$ | $\Delta(t_{2},t_{3})$ | $\Delta_{c \rightarrow i}$ | $\Delta_{i \rightarrow c}$ |
> > | :---: | :---: | :---: | :---: | :---: | :---: | :---: | :---: |
> > | 3.1-8B | 52.2 | 19/22 | 61.0 | 61.2 | 0.2 | 0/3 | 1/18 |
> > | 3.1-70B | 65.8 | 0 | 77.4 | 77.4 | 0 | \- | \- |
> > | 3.3-70B | 75.6 | 4/24 | 77.8 | 77.8 | 0 | \- | \- |
> >
> >
> > **Per-turn performance analysis: all turns, 8B**
> > | Turn $l$ | verif.acc.@$t_{l}$ | acc.@$t_{l}$ | acc.@$t_{l+1}$ | $\Delta(t_{l},t_{l+1})$ | $\Delta_{c \rightarrow i}$ | $\Delta_{i \rightarrow c}$ |
> > | :---: | :---: | :---: | :---: | :---: | :---: | :---: |
> > | 1 | 401/500 | 59.0 | 61.0 | 2.0 | 8/29 | 18/79 |
> > | 2 | 19/22 | 61.0 | 61.2 | 0.2 | 0/3 | 1/18 |
> > | 3 | 6/8 | 61.2 | 61.0 | \-0.2 | 2/2 | 1/6 |
> > | 4 | 2/2 | 61.0 | 61.0 | 0.0 | \- | 0/2 |
> > | 5 | 1/1 | 61.0 | 61.0 | 0.0 | \- | 0/1 |
> > | 6 | 0/1 | 61.0 | \- | \- | \- | \- |
> >
> >
> > ---
> > **References:**
> >
> > [1] Kumar, Aviral, et al. "Training language models to self-correct via reinforcement learning." arXiv preprint arXiv:2409.12917 (2024).
> >
> > [2] Marjanović, Sara Vera, et al. "DeepSeek-R1 Thoughtology: Let's think about LLM Reasoning." arXiv preprint arXiv:2504.07128 (2025).
> >
> > [3] Ma, Wenjie, et al. "Reasoning Models Can Be Effective Without Thinking." arXiv preprint arXiv:2504.09858 (2025).
> >
> > [4] Chen, Jiefeng, et al. "SETS: Leveraging Self-Verification and Self-Correction for Improved Test-Time Scaling." arXiv preprint arXiv:2501.19306 (2025).
> >
> > [5] Qu, Yuxiao, et al. "Recursive introspection: Teaching language model agents how to self-improve." Advances in Neural Information Processing Systems 37 (2024): 55249-55285.

---

> ### Comment · Reviewer_VSGv · 2025-06-05
> **Response to authors**
>
> Hi,
>
> I thank the authors for their diligent rebuttal.
>
> 1. Self-correction turns
> - It is good to see that the model is not limited to 1 or 2 turns during inference even though the training (SFT) data subjects it to limited turns. I think this point is nuanced and not properly mentioned in the paper which led to the confusion.
> - It is interesting to see that 8B and 70B models go uptill 6 and 3 turns, respectively, and the revised paper should include those numbers.
>
> 2. Thinking models
> - I agree that the reward modeling on final answer correctness as well as verification steps can provide potentially useful signals; and quite unique to the design of this approach.
>
> 3. Related work and presentation
> - Sounds good!
>
> Having gone through other reviews, I do agree that the paper will benefit from inclusion of more models and more popular RL algorithms like GRPO in the future versions. I have increased my score to 7 to reflect my updated opinion.

---

> > ### Author Response · Authors · 2025-06-05
> >
> > Thank you very much for your encouraging feedback! We will incorporate all responses in the updated version of the paper.

---

### Official Review · Reviewer_MWdF · 2025-05-15

**Rating:** 4
**Confidence:** 3
**Ethics Flag:** 1

**Summary:**

This paper introduces SPOC (Spontaneous Self-Correction), a novel approach for enhancing mathematical reasoning in large language models (LLMs). Unlike existing self-correction methods that rely on external prompts or post-generation refinement, SPOC enables LLMs to interleave solution generation and verification in a single inference pass, dynamically terminating based on verification outcomes. By framing the process as a multi-agent interaction (solution proposer and verifier) within the same model, SPOC uses synthetic data for fine-tuning and online reinforcement learning (RL) with rewards for both solution correctness and verification accuracy. Experiments on benchmarks like MATH500, AMC23, and AIME24 show significant improvements: Llama-3.1-8B/70B achieve gains of 8.8%/11.6% on MATH500, 10.0%/20.0% on AMC23, and 3.3%/6.7% on AIME24, respectively. The RLOO RL variant further boosts performance, reaching 94.6% accuracy on DeepSeek-R1-Distill-Llama-70B for MATH500.

**Questions To Authors:**

N/A

**Reasons To Accept:**

- SPOC outperforms baseline methods across model sizes and task difficulties
- By modeling the LLM as both a solution proposer and verifier, SPOC introduces a self-play training strategy that avoids the need for separate models or stronger "teacher" supervision

**Reasons To Reject:**

- SPOC with RLOO performs much better than RAFT, so it makes more sense to use RLOO as the default RL algorithm and compare it with RLOO as the baseline.
- SPOC relies on synthetic data generated by the base model for fine-tuning, which may introduce bias if the initial model has systematic errors.
- SPOC only integrates with RAFT and RLOO. It is not clear whether it will be compatible with other RL algorithms.

---

> ### Author Response · Authors · 2025-06-03
> **Response to Reviewer MWdF**
>
> Thanks for your feedback and the time you have taken. We provide detailed clarifications on your concerns as follows.
>
> > **Q1 Self-improvement.** SPOC relies on synthetic data generated by the base model for fine-tuning, which may introduce bias if the initial model has systematic errors.
>
> **A1** Self-improvement approaches finetune the base model with fully self-synthesized data, serving as an important complement of distillation when expert distillation data is often unavailable or expensive in practice. To mitigate potential bias embedded in initial models, we conduct extensive experiments covering a diverse range of initial model capacities. In addition to widely-used Llama models, we also experimented on DeepSeek-R1-Distill models (both 8B & 70B) which are distilled from DeepSeek-R1 with modified configs and tokenizers [3]. Such experimental extensiveness promotes sufficient coverage of initial model capabilities and reasoning levels, and our results demonstrate consistent effectiveness and generalizability across initial models and task difficulties.
>
>
> > **Q2 Policy optimizer.** SPOC with RLOO performs much better than RAFT, so it makes more sense to use RLOO as the default RL algorithm and compare it with RLOO as the baseline.
>
> **A2** We aim to showcase the improvement achieved by SPOC’s reflective reasoning, rather than merely reaching a better performance using a stronger baseline optimizer. Empirically, SPOC with RAFT reaches consistent performance boosts across base models and task difficulties, as shown in Table 1. RAFT’s robustness allows for wide applications over diverse learning scenarios [1,2]. Our extensive experiments promote proper and fair interpretations of SPOC’s enhancements and robustness.
>
> We apply both RAFT and RLOO to provide a comprehensive understanding, as they follow different optimization schemes, where the former selects the best-of-N response while the latter uses all generated responses to optimize the policy. Consistent and significant performance improvements shown in our results demonstrate our method’s effectiveness.
>
>
> > **Q3 Policy optimizer.** SPOC only integrates with RAFT and RLOO. It is not clear whether it will be compatible with other RL algorithms.
>
> **A3** As mentioned above, RAFT and RLOO are representative multi-rollout algorithms following different optimization schemes. Considering single-rollout as a special case of multi-rollout (K=1), other single-rollout algorithms (eg. REINFORCE or PPO or extended variants) also naturally fit our learning formulation. Our learning formulation and empirical experiments demonstrate SPOC’s general compatibility with any RL algorithm.
>
> ---
> **References:**
>
> [1] Xiong, Wei, et al. "Self-rewarding correction for mathematical reasoning." arXiv preprint arXiv:2502.19613 (2025).
>
> [2] Xu, Tengyu, et al. "The perfect blend: Redefining RLHF with mixture of judges." arXiv preprint arXiv:2409.20370 (2024).
>
> [3] Guo, Daya, et al. "Deepseek-r1: Incentivizing reasoning capability in llms via reinforcement learning." arXiv preprint arXiv:2501.12948 (2025).

---

> > ### Author Response · Authors · 2025-06-10
> > **We appreciate your valuable time and reviews**
> >
> > Dear Reviewer MWdF,
> >
> > We sincerely appreciate your feedback and the time you have dedicated. As the discussion period concludes soon (June 10th), please let us know if you have any questions or concerns. We would be delighted to provide any additional clarifications.
> >
> > Best Regards,
> >
> > Authors of 1455

---

### Official Review · Reviewer_VrYp · 2025-05-17

**Rating:** 5
**Confidence:** 4
**Ethics Flag:** 1

**Summary:**

The paper proposes SPOC (Spontaneous Self-Correction), a pipeline that teaches a single model to alternate between proposer and verifier roles within one forward pass. Synthetic “solution + verification” dialogues are first generated with Pair-SFT; the policy is then refined via message-level RL (RAFT or RLOO) using a strict 0/1 reward that requires both a correct answer and a validated proof.

The writing is generally clear and the figures convey the workflow well. Ablations on reward design and iteration depth help justify key choices. Still, section 3 is notation-heavy for a paper without formal theory, trimming some symbols or moving details to an appendix would improve readability.

Empirically, SPOC consistently lifts several Llama-family backbones and their DeepSeek-distilled variants on MATH500, AMC 2023 and AIME 2024. To strengthen the method's applicability to other model families, it would be helpful to show results on a different architecture (e.g., Qwen 2.5) or at least discuss why the method should transfer beyond Llama, especially that Qwen has been a popular base model to train for long CoT and reflective reasoning.

In terms of related work discussion, [1-2] are very relevant in terms of relying on the conversations of multiple role-playing LLMs (proposer, verifier, ect.) for synthetic data curation and further distillation. In the attempts to reproduce o1 thinking, there have also been works discussion on how to construct long CoTs (self-reflection, ect.) for SFT and later RL [3]. The concurrent line of work on distillation is also highly relevant [4-5], though I understand that these are quite recent.

In general, the manuscript would benefit from clarifying what is the main advantage of SPOC. Moreover, multi-agent data generation causes extra latency as compared to distillation from a strong model or simply forcing reflective keyword such as "wait" during rollouts or inference time for data collection.

[1] Flow-DPO: Improving LLM Mathematical Reasoning through Online Multi-Agent Learning

[2] Synthesizing Post-Training Data for LLMs through Multi-Agent Simulation

[3] O1 Replication Journey: A Strategic Progress Report – Part 1

[4] s1: Simple test-time scaling

[5] LIMO: Less is More for Reasoning

**Reasons To Accept:**

- Overall, the work is technically solid and the empirical results showed notable improvements.
- The paper is mostly well written and its diagrams easy to follow. Ablation studies provide interesting insights.

**Reasons To Reject:**

- The experiments are done within the same model line (Llama).
- The concept of first curating data with reflective reasoning behaviors and distilling it via SFT or RL is not quite new. I'd be curious to learn the unique finding of this paper.
- Multi-turn (agent) conversation naturally has a high latency cost as compared to direct distillation from strong models or simply forcing reflections by appending words like "wait" or "let me double check" at inference time (or rollouts) for data construction. It's not clear why or on what scenario would this approach be more beneficial.
- Math notations are heavy and some unnecessary, given that the paper does not present theoretical analysis.

---

> ### Author Response · Authors · 2025-06-03
> **Response to Reviewer VrYp (1/2)**
>
> We appreciate your insightful feedback and the time you have dedicated. Please kindly find our responses to each point as follows.
>
> > **Q1 Model family.** The experiments are done within the same model line (Llama).
> >
> >  ... at least discuss why the method should transfer beyond Llama ...
>
> **A1** Despite the Llama family, our experiments cover different model capacities and parameter scales (especially 70B full-finetuning often lacking in literature). Our experiments also include DeepSeek-R1-Distill models (both 8B & 70B) which are distilled from DeepSeek-R1 with modified configs and tokenizers [6]. Due to different parameter scales, training data, and training approaches, base models covered in our work exhibit diverse reasoning capabilities and behaviors. Since SPOC demonstrates consistent effectiveness and generalizability across a broad spectrum of initial models and task difficulties, our empirical results strengthen SPOC's applicability to other model families.
>
> > **Q2 Related work and main advantage.** The concept of first curating data with reflective reasoning behaviors and distilling it via SFT or RL is not quite new. I'd be curious to learn the unique finding of this paper.
> >
> > In general, the manuscript would benefit from clarifying what is the main advantage of SPOC.
>
> **A2** Compared to recent literature, SPOC showcases the following unique advantages:
> 1. **Simple mechanism design and dynamic reflective reasoning.** Our training pipeline effectively bootstraps the model’s spontaneous self-verification and correction, enabling explicit signaling of continuation/termination, thereby adaptive generation.
> 2. **Joint optimization of both solutions and verifications.** The solution outcome reward does not supervise intermediate messages within a multi-turn generation before the final answer. Besides outcome reward, SPOC applies process supervision to reward correct and penalize incorrect *solutions and verifications*. Ablations underscore the importance of such process-level supervision.
> 3. **Consistent enhancement across initial models and task difficulties.** Extensive experiments demonstrate SPOC’s effectiveness and generalizability as highlighted earlier.
>
> We appreciate the reviewer for bringing up related works. We will incorporate the following discussions in our revised manuscript:
>
> 1. [1-2] utilize multi-agent conversations. Similar to MCTS-DPO [7], [1] focuses on constructing preference pairs with shared prefix for preference optimization, rather than enabling reflective reasoning. [2] synthesize training data via social scenario simulations which involves complicated mechanism design and numerous instruction prompting, whereas SPOC relies on much simpler pipeline and minimal prompting.  Besides, [2] distill from DeepSeek-R1-Distill-Qwen-32B for reasoning tasks, while SPOC is a *self-improvement* approach that fine-tunes initial models with fully self-synthesized data.
> 2. Recent works on long COTs have emerged [3,6] as an effective approach. Beyond outcome rewards, SPOC provides message-wise supervision to guide learning. Moreover, SPOC’s multi-turn formalism enables compatibility with R1’s long thinking, demonstrated by experiments on DeepSeek-R1-Distill models.
> 3. [4-5] leverages distillation on small datasets, with training data curated using strong models and multiple selection criteria. [4] proposes budget forcing to enforce sequential test-time scaling, while [5] investigates the impact of prerequisite knowledge. Both works complement the line of self-improvement approaches.
>
> *(continued next)*

---

> > ### Author Response · Authors · 2025-06-03
> > **Response to Reviewer VrYp (2/2)**
> >
> > *(continuing)*
> >
> > > **Q3 High latency.** Multi-turn (agent) conversation naturally has a high latency cost as compared to direct distillation from strong models or simply forcing reflections by appending words like "wait" or "let me double check" at inference time (or rollouts) for data construction.
> >
> > **A3** Although direct distillation also achieves strong performance, distillation is not fairly comparable to SPOC due to its self-improvement nature, i.e., SPOC fine-tunes initial models with fully self-synthesized data. Self-improvement is applicable to a broad range of real-world scenarios, especially when expert distillation data is unavailable or expensive. Moreover, distillation and self-improvement are not mutually exclusive. Distilled models could further apply self-improvement methods, established by [6] and our experiments on DeepSeek-R1-Distill models.
> >
> > Keyword forcing triggers reflections by appending keywords like “wait” at inference time [4]. Keyword forcing relies on token budgets, and the amount and timing of reflections are rather predefined hyperparameters. Differently, SPOC spontaneously elicits reflections, and adapts solution generation based on verification correctness, thereby achieving better inference flexibility. Moreover, SPOC applies process supervision to reward correct and penalize incorrect *solutions and verifications*. In contrast, learning with keyword-forced rollouts and outcome rewards does not supervise intermediate reflective reasoning steps within the long response, potentially resulting in undesirable excessively long reasoning [8]. As noted in Section 5, to address the prohibitive length of long CoTs, an interesting future direction is to extend SPOC to partial solutions in long CoTs, using step-level process rewards to guide RL training and enable dynamic revisions when errors are detected until reaching the final answer.
> >
> > > **Q4 Math notations.** Math notations are heavy and some unnecessary, given that the paper does not present theoretical analysis.
> >
> > **A4** Our math notations are mainly used to formulate the multi-agent paradigm, where the analysis for Nash equilibrium is omitted due to its simplicity. We will condense Section 3 and move details to the appendix in our revised manuscript.
> >
> > ---
> > **References:**
> >
> > [1] Flow-DPO: Improving LLM Mathematical Reasoning through Online Multi-Agent Learning
> >
> > [2] Synthesizing Post-Training Data for LLMs through Multi-Agent Simulation
> >
> > [3] O1 Replication Journey: A Strategic Progress Report – Part 1
> >
> > [4] s1: Simple test-time scaling
> >
> > [5] LIMO: Less is More for Reasoning
> >
> > [6] Guo, Daya, et al. "Deepseek-r1: Incentivizing reasoning capability in llms via reinforcement learning." arXiv preprint arXiv:2501.12948 (2025).
> >
> > [7] Xie, Yuxi, et al. "Monte carlo tree search boosts reasoning via iterative preference learning." arXiv preprint arXiv:2405.00451 (2024).
> >
> > [8] Marjanović, Sara Vera, et al. "DeepSeek-R1 Thoughtology: Let's think about LLM Reasoning." arXiv preprint arXiv:2504.07128 (2025).

---

> > > ### Author Response · Authors · 2025-06-10
> > > **We appreciate your time and valuable feedback**
> > >
> > > Dear Reviewer VrYp,
> > >
> > > We sincerely appreciate your efforts in providing thoughtful reviews. As the discussion period concludes soon (June 10th), please let us know if you have any questions or concerns. We would be delighted to provide any additional details or clarifications.
> > >
> > > Best Regards,
> > >
> > > Authors of 1455

---

> > > > ### Comment · Reviewer_VrYp · 2025-06-11
> > > > **Thank you for the rebuttal**
> > > >
> > > > Thank the authors for their detailed responses. My concern remains on the novelty/uniqueness of SPOC.
> > > >
> > > > - While I appreciate the simplicity and effectiveness of SPOC, the arguments presented do not sufficiently clarify the specific scenarios where SPOC would be preferable to existing methods.
> > > > - Specifically, I'm not sure about the characterization of the reward as "process-based" [1]. As the agent receives feedback only after completing the entire task (either problem-solving or verification), it functions more as an outcome-based reward, similar to standard approaches. The reward density would not be significantly higher than in other methods, especially for interactions with just a few turns.
> > > >
> > > > Furthermore, while SPOC's self-improvement capability is a valid point, the accessibility of powerful open-source models for distillation at a low cost remains a strong alternative. It's unclear why one would choose SPOC over distillation, particularly when options like the R1 API are readily and affordably available. If we consider the scenario of training the strongest models (e.g. no better model for distillation), then cold starting with RL is sufficient to self-improve as demonstrated by R1.
> > > >
> > > > To strengthen the paper, I suggest a deeper analysis of the unique benefits of curating data from multi-LLM conversations. Investigating whether this approach elicits more desirable emergent behaviors compared to other data sources could provide a compelling argument for SPOC's novelty.
> > > >
> > > > I acknowledge the delay of my reply in the rebuttal period and will not object to acceptance if the other reviewers have reached a consensus.
> > > >
> > > > [1] Let's Verify Step by Step

---

> ### Author Response · Authors · 2025-06-11
> **Thank you for the follow-up feedback**
>
> We highly appreciate the reviewer’s follow-up feedback. Please kindly find our further clarifications as follows.
>
> > **Q5 Application scenarios.** While I appreciate the simplicity and effectiveness of SPOC, the arguments presented do not sufficiently clarify the specific scenarios where SPOC would be preferable to existing methods.
>
> **A5** Self-correction is a desirable capability that enables the model to improve over initial responses.  SPOC applies to scenarios where (1) self-correction is beneficial for reasoning tasks, (2) *agent use cases that require multi-turn execution*, and (3) *adaptive response generation efficiently utilizes computational cost at inference time*. Specifically for point (3), long-thinking models have excessive thought length [8], resulting in high computational costs at deployment and performance degradation. SPOC seeks to improve reasoning performance in the meantime achieving *efficient* test-time scaling. Per-turn performance analysis (table 2 and tables in responses to Reviewer VSGv) suggest that stronger models tend to **achieve correct solutions sooner**, without trying many additional self-corrections. This observation aligns with our discussion in Section 4.2.
>
> Moreover, we’d like to clarify that SPOC is not intended to replace existing methods. In fact, SPOC is well compatible with R1’s long thinking, demonstrated by experiments on DeepSeek-R1-Distill models. These experiments support our accessesment of SPOC’s effectiveness and generalizability across initial model capabilities. Furthermore, the multi-turn formalism serves as an exploration towards reflective reasoning within long CoT. As noted in Section 5, to address the undesiarable prohibitive length, an interesting future direction is to extend SPOC to partial solutions within long CoTs, using step-level process rewards to guide RL training and enable dynamic revisions when errors are detected until reaching the final answer.
>
> > **Q6 Process feedback.** I'm not sure about the characterization of the reward as "process-based"...
>
> **A6** We appreciate the reviewer for bringing up [1], which characterizes a *step-level* process reward. To clarify, our notion of process supervision is at the *message level* (illustrated in Figure 3a), as compared to the outcome reward solely on the correctness of the entire trajectory’s final answer (illustrated in Figure 3c). Figure 3c characterizes a setting where the entire trajectory is considered correct if the final answer is correct, regardless of potentially incorrect intermediate verifications/solutions. Such reward does not penalize unnecessarily long trajectories, and is not as effective as message-level supervision demonstrated by our ablations.
>
> Concurrent work on self-correction [2] adopts a similar description. We will include the clarification above in the revised version of our manuscript.
>
>
> **Q7 Self-improvement and distillation.** It's unclear why one would choose SPOC over distillation, particularly when options like the R1 API are readily and affordably available. If we consider the scenario of training the strongest models (e.g. no better model for distillation), then cold starting with RL is sufficient to self-improve as demonstrated by R1.
>
> **A7** We’d like first to clarify that this work does not advocate choosing SPOC over distillation. Distillation and self-improvement are well compatible with each other. Our experiments on DeepSeek-R1-Distill models fit the scenario of applying both methods. Obtained by distilling from R1, these models already achieve highly strong results over reasoning tasks, and SPOC further boosts their performance consistently.
>
> Also, this paper is not intended to replicate R1, as it was concurrently developed. We mainly aim to investigate the benefit of **dynamic spontaneous self-corrections**, and introduce a learning framework to initialize such behavior and improve reasoning capabilities.
>
> ---
> **References:**
>
> [2] Ma, Ruotian, et al. "S $^ 2$ R: Teaching LLMs to Self-verify and Self-correct via Reinforcement Learning." arXiv preprint arXiv:2502.12853 (2025).

---

### Official Review · Reviewer_Cxxp · 2025-05-21

**Rating:** 6
**Confidence:** 4
**Ethics Flag:** 1

**Summary:**

The paper proposes SPOC (Spontaneous Self-Correction), a training-and-decoding framework that interleaves a solution proposer and a verifier within a single inference pass. The authors first generate synthetic “solution + verification” pairs (Pair-SFT) to teach the model this format, and then apply online reinforcement learning with a reward that jointly incentivizes correct solutions and correct verifications. Experiments on three math-reasoning benchmarks (MATH500, AMC23, AIME24) across multiple Llama-3 and DeepSeek models show consistent accuracy gains.

**Questions To Authors:**

1. Could you provide a per-turn performance analysis, similar to Table 2 for MATH500, for the AMC23 and AIME24 benchmarks to better understand SPOC's generalizability?

2. Beyond the reported Verif.Acc.@t1, were the verifier's false positive and false negative rates measured? Additionally, was the verifier's behavior on slightly incorrect explanations examined, and could you share any related statistics such as a confusion matrix or an error-type breakdown?

**Reasons To Accept:**

**Clarity and Organization.** The writing is clear and easy to follow, and the paper’s structure is well organized.

**Integrated Self-Correction.** SPOC embeds solution generation and verification into a single inference run, eliminating external prompt loops and simplifying deployment.

**Clear Two-Stage Pipeline.** The synthetic Pair-SFT stage followed by online RL is precisely defined and delivers substantial accuracy gains across both 8B and 70B models on multiple benchmarks.

**Reasons To Reject:**

**Limited Per-Turn Analysis.** Table 2 provides a detailed turn-by-turn breakdown only for MATH500; the same analysis is missing for AMC23 and AIME24, making it difficult to assess SPOC’s effectiveness across all datasets.

**High Computational Cost.** The online RL phase requires 32 × H100 GPUs (Appendix B), which is prohibitively expensive for most research groups and small teams.

**Incomplete Verifier Reliability Reporting.** Although the paper reports a single Verif.Acc.@t1 metric (e.g., approximately 80 % on MATH500), it does not provide deeper diagnostics such as false-positive/false-negative rates, confusion matrices, or robustness to subtly incorrect justifications, leaving the verifier’s reliability only partially evaluated.

---

> ### Author Response · Authors · 2025-06-03
> **Response to Reviewer Cxxp**
>
> We appreciate your constructive comments and the time you have taken. Your valuable suggestions have significantly elevated our work. Please kindly find our detailed response below.
>
> > **Q1 Verifier Reliability.** Beyond the reported Verif.Acc.@t1, were the verifier's false positive and false negative rates measured? Additionally, was the verifier's behavior on slightly incorrect explanations examined, and could you share any related statistics such as a confusion matrix or an error-type breakdown?
>
> **A1** We provide more detailed diagnostics in the table down below. Each confusion matrix corresponds to a base model and task pair, with the rows and columns indicating the actual and predicted solution correctness, respectively — i.e., diagonal cells represent the true positive (TP) and true negative (TN) rates while the off-diagonal cells represent the false positive (FP) and false negative (FN) rates. We observe the following phenomena:
> 1. On easier tasks, the proposer has higher solution accuracy, and the verifier tends to show higher TP&FP and lower TN&FN.
> 2. Stronger models that reach higher solution accuracy also have higher TP&FP.
> 3. The small model’s high verification accuracy attributes largely to its higher TN.
>
> We will include the table in the appendix of our revised manuscript.
>
> > **Q2 Per-Turn Performance Analysis.** Could you provide a per-turn performance analysis, similar to Table 2 for MATH500, for the AMC23 and AIME24 benchmarks to better understand SPOC's generalizability?
>
> **A2** We provide the per-turn performance statistics for AMC23 and AIME24 as follows. The results are consistent with MATH500 analysis in Table 2. SPOC generally improves or maintains performance on the second solution turns. The smaller model has lower final accuracy yet larger turn-wise improvements, while larger models tend to achieve correct solutions sooner at turn1. Moreover, turn-wise corrections occurs less in these two challenging competition benchmarks, as they contain significantly fewer questions than MATH500. We will include both tables in the appendix of our revised manuscript.
>
> > **Q3 Computational Cost.** The online RL phase requires 32 × H100 GPUs (Appendix B), which is prohibitively expensive for most research groups and small teams.
>
> **A3** We utilized stated compute to accelerate experiments across various initial models and parameter scales (especially 70B full-finetuning often lacking in the literature), so as to comprehensively evaluate our method’s effectiveness and generalizability. Our results not only demonstrate SPOC’s efficacy across model scales, but also present interesting different behaviors by different model sizes (such as observations on FP&FN above), promoting a better understanding of SPOC’s strengths.
>
> Each 8B run takes approximately 8~12 hours with the stated compute. If the runtime and experimentation on large models are not a concern, 4 x H100 GPUs are sufficient to conduct the RL training.
>
> ---
>
> **Verifier reliability performance**
> | Base Model | MATH500 |  | AMC2023 |  | AIME2024 |  |
> | :---: | :---: | :---: | :---: | :---: | :---: | :---: |
> | 3.1-8B | 90.2 (266/295) | 9.8 (29/295) | 81.9 (9/11) | 18.2 (2/11) | 0 (0/1) | 100 (1/1) |
> |  | 34.1 (70/205) | 65.9 (135/205) | 24.1 (7/29) | 75.9 (22/29) | 0 (0/29) | 100 (29/29) |
> | 3.1-70B | 100 (385/385) | 0 (0/385) | 100 (21/21) | 0 (0/21) | 85.7 (6/7) | 14.3 (1/7) |
> |  | 87.0 (100/115) | 13.0 (15/115) | 84.2 (16/19) | 15.8 (3/19) | 82.6 (19/23) | 17.4 (4/23) |
> | 3.3-70B | 99.0 (385/389) | 1.0 (4/389) | 93.1 (27/29) | 6.9 (2/29) | 100 (7/7) | 0 (0/7) |
> |  | 78.4 (87/111) | 21.6 (24/111) | 72.7 (8/11) | 27.3 (3/11) | 82.6 (19/23) | 17.4 (4/23) |
>
> **Per-turn performance analysis**
>
> AIME 2024
> | Base model | base.acc. | verif.acc.@$t_1$ | acc.@$t_1$ | acc.@$t_2$ | $\Delta(t_{1},t_{2})$ | $\Delta_{c \rightarrow i}$ | $\Delta_{i \rightarrow c}$ |
> | :---: | :---: | :---: | :---: | :---: | :---: | :---: | :---: |
> | 3.1-8B | 3.3 | 29/30 | 1/30 | 2/30 | 1/30 | 0/1 | 1/7 |
> | 3.1-70B | 16.7 | 10/30 | 7/30 | 7/30 | 0/30 | 0/1 | 0/1 |
> | 3.3-70B | 26.7 | 11/30 | 7/30 | 7/30 | 0/30 | 0 | 0/1 |
>
> AMC 2023
> | Base model | base.acc. | verif.acc.@$t_1$ | acc.@$t_1$ | acc.@$t_2$ | $\Delta(t_{1},t_{2})$ | $\Delta_{c \rightarrow i}$ | $\Delta_{i \rightarrow c}$ |
> | :---: | :---: | :---: | :---: | :---: | :---: | :---: | :---: |
> | 3.1-8B | 22.5 | 31/40 | 11/40 | 13/40 | 5.0 | 0/2 | 2/11 |
> | 3.1-70B | 32.5 | 24/40 | 21/40 | 21/40 | 0 | 0 | 0 |
> | 3.3-70B | 57.5 | 30/40 | 29/40 | 28/40 | \-2.5 | 1/2 | 0/2 |

---

> > ### Comment · Reviewer_Cxxp · 2025-06-06
> >
> > I really appreciate the careful responses by the authors. My concerns are well resolved; therefore, I will keep my original rating   leaning toward the acceptance.

---

> > > ### Author Response · Authors · 2025-06-07
> > >
> > > We really appreciate your recognition of our contributions and encouraging feedback! We will incorporate all responses in the revised version of our manuscript.

---

### Official Review · Reviewer_Vfpt · 2025-05-25

**Rating:** 5
**Confidence:** 4
**Ethics Flag:** 1

**Summary:**

This paper proposes SPOC, a training method that allows LLMs to interleave solution generation and verification in order to improve math reasoning abilities of an array of SLMs. The paper differs from prior work as it interleaves the model's ability to generate solutions and perform self-verification in the same reasoning trace, instead of separate prompts. Their method combines pair-wise SFT strategy to enable multi-turn (solution and verification) generation, followed by an online RL training phase (RAFT or RLOO).

**Questions To Authors:**

See weaknesses

**Reasons To Accept:**

- The paper is fairly well written and easy to follow
- Empirically strong results and improvements on math reasoning tasks with a variety of SLMs show that their methodology is effective

**Reasons To Reject:**

- The paper lacks a GRPO baseline which would test the model could learn to interleave solving and verifying with online RL on the same data -- without which it is hard to gauge their method's effectiveness.
- The main contribution is *not* properly placed in recent literature, specifically, it is unclear how SPOC compares to other long CoT methods, which have also demonstrated interleaved solving, backtracking, and verification in longer CoT generations (see below):
     - https://arxiv.org/abs/2503.01307
     - https://arxiv.org/abs/2504.07128

---

> ### Author Response · Authors · 2025-06-03
> **Response to Reviewer Vfpt**
>
> Thank you for your insightful feedback and the time you have taken. Please kindly find our detailed responses below.
>
> > **Q1** **Policy optimizer.** The paper lacks a GRPO baseline which would test the model could learn to interleave solving and verifying with online RL on the same data -- without which it is hard to gauge their method's effectiveness.
>
> **A1** We would like to highlight that our message-level RL framework is compatible with any policy optimization method. We aim to showcase the improvement achieved by SPOC’s reflective reasoning, rather than merely reaching a better performance using a stronger baseline optimizer. To provide a comprehensive understanding we apply RAFT and RLOO, two representative algorithms following different optimization schemes, where the former optimizes over the best-of-N response while the latter optimizes over all generated responses. Our experiments have shown that SPOC with either optimizer achieves consistent and significant performance improvements across model capacities and benchmark difficulties, demonstrating our method’s effectiveness.
>
> GRPO is conceptually very similar to RLOO, as they rely on similar advantage update to provide process supervision for every message relative to the others [1]. Since our results already show that RLOO yields stronger results than RAFT, our experiments present sufficient evidence to demonstrate the benefits of the advantage function.
>
> > **Q2 Related work.** it is unclear how SPOC compares to other long CoT methods, which have also demonstrated interleaved solving, backtracking, and verification in longer CoT generations
>
> **A2** We appreciate the reviewer for bringing up related works. We will include discussions below to updated manuscript.
>
> Gandhi et al. [2] investigate how priming with controlled behavioral datasets affects subsequent RL training, where such datasets are either curated by Claude-3.5-Sonnet on Countdown or by Qwen-2.5-32B from OpenWebMath.
> In contrast to distillation from stronger models, SPOC is a *self-improvement* approach that fine-tunes initial models with self-generated synthetic data throughout the training pipeline.
>
> Marjanovic et al. [3] conduct a systematic investigation of DeepSeek-R1’s reasoning processes. Compared to R1’s long CoTs, SPOC employs a spontaneous reasoning strategy that adapts  generations based on verification outcomes. Beyond R1’s solution outcome reward, SPOC applies process supervision to reward correct and penalize incorrect *solutions and verifications*.
> Moreover, SPOC’s multi-turn formalism is compatible with R1’s long CoT, demonstrated by experiments on DeepSeek-R1-Distill models (both 8B & 70B) in Table 1. SPOC shows consistent effectiveness and generalizability across initial model capacities and task difficulties.  As noted in Section 5, to address the prohibitive length of long CoTs [3], an interesting future direction is to extend SPOC to partial solutions in long CoTs, using step-level process rewards to guide RL training and enable dynamic revisions when errors are detected until reaching the final answer.
>
>
> ---
> **References:**
>
> [1] Lambert, Nathan. "Reinforcement Learning from Human Feedback." arXiv preprint arXiv:2504.12501 (2025).
>
> [2] Gandhi, Kanishk, et al. "Cognitive behaviors that enable self-improving reasoners, or, four habits of highly effective stars." arXiv preprint arXiv:2503.01307 (2025).
>
> [3] Marjanović, Sara Vera, et al. "DeepSeek-R1 Thoughtology: Let's think about LLM Reasoning." arXiv preprint arXiv:2504.07128 (2025).

---

> > ### Comment · Reviewer_Vfpt · 2025-06-09
> >
> > Thank you for the clarification. I would like to see this discussion of the related work in the paper.

---

> > > ### Author Response · Authors · 2025-06-09
> > >
> > > Thank you very much for your feedback. As we are unable to upload a revised PDF during the rebuttal period, we'd like to reconfirm we will include discussions under **Q2 Related work** in the updated manuscript.
> > >
> > > Please let us know if you have any other questions or concerns. We would be delighted to provide any additional clarifications.

---

### Decision · Program_Chairs · 2025-07-08

**Decision:**

Accept

**Comment:**

The study proposes SPOC, which is a process that allows an LLM to generate interleaved solution+verification messages in a single forward pass. This is trained using message-level rewards via RAFT/RLOO. The reviews were mixed for this submission, trending negative, with several concerns and strengths raised. The main ones I'll highlight that were common across multiple reviews are:

Strengths:

* Strong results across multiple math datasets showing the effectiveness of SPOC.
* Technical novelty of "closed-loop"/single-pass generation and verification.

Weaknesses:

* More RL baselines (such GRPO) in addition to RLOO
* Some presentation details were not clear, or mathematical notation is too heavy.

Additional concerns or strengths that are not as pertinent to the scientific contribution of SPOC include and were either addressed by the authors or only raised as a concern by one reviewer include: missing related work (addressed in rebuttal), concerns about the number of turns in data/analysis (addressed in rebuttals), limitation to Llama-derived models instead of Qwen-series, computational cost, or using synthetic data for fine-tuning.

Ultimately, I'm convinced the contribution and novelty outweighs missing RL-related baselines, especially since RLOO and GRPO are similar methods and +RLOO is not only contribution in this work (although that is what leads to the best performance; the conclusions would not be different if +GRPO was the winner): This paper can be presented at COLM.